# PRIMUS: ENFORCING ATTENTION USAGE FOR 3D MEDICAL IMAGE SEGMENTATION

## ABSTRACT

Transformers have achieved remarkable success across multiple fields, yet their impact on 3D medical image segmentation remains limited with convolutional networks still dominating major benchmarks. In this work, a) we analyze current Transformer-based segmentation models and identify critical shortcomings, particularly their over-reliance on convolutional blocks. Further, we demonstrate that in some architectures, performance is unaffected by the absence of the Transformer, thereby demonstrating their limited effectiveness. To address these challenges, we move away from hybrid architectures and b) introduce Transformer-centric segmentation architectures, termed Primus and PrimusV2. Primus leverages high-resolution tokens, combined with advances in positional embeddings and block design, to maximally leverage its Transformer blocks, while PrimusV2 expands on this through an iterative patch embedding. Through these adaptations, Primus surpasses current Transformer-based methods and competes with a default nnU-Net while PrimusV2 exceeds it and is on par with the state-of-the-art ResEnc-L architecture across nine public datasets. In doing so, we introduce the first competitive Transformer-centric model, making Transformers state-of-the-art in 3D medical segmentation. Our code will be published.

## 1 INTRODUCTION

The success of the attention mechanism and the Transformer architecture (Vaswani et al., 2017) initiated a paradigm shift in natural language processing, computer vision and several other domains (Chang et al., 2023; Ahmed et al., 2023; Aleissaee et al., 2023; Khan et al., 2022; Zhang et al., 2023). The domain of medical image segmentation has been no exception, also experiencing a large influx of Transformer-based architectures for 3D medical image segmentation, one of the most significant tasks in medical image analysis (Appendix D.1). With the ability of Transformers to learn long-range dependencies (Vaswani et al., 2017; Zimerman & Wolf, 2024), it was believed that incorporating Transformers would enable architectures to learn global patterns that convolutional neural network (CNN) architectures could not. Despite this promise and many efforts of replacing convolutions with attention in medical image segmentation (Zhang et al., 2025), multiple large-scale evaluations have demonstrated the inability of current Transformers to outperform CNN architectures (Isensee et al., 2024; Bassi et al., 2024).

This lack of performance in the 3D medical image segmentation domain is not necessarily surprising, given the well-known difficulties of training Transformers from scratch (Dosovitskiy et al., 2021; Touvron et al., 2021a). This is particularly severe in a domain where the majority of supervised datasets are in the order of hundreds of samples (Litjens et al., 2017) and not millions (Appendix D.2), and where training from scratch with dynamically planned CNN architectures (Isensee et al., 2021) is still commonplace. Irrespective of these roadblocks, a plethora of Transformer architectures have been proposed (Cao et al., 2022; Hatamizadeh et al., 2021; 2022; Zhang et al., 2021; Wang et al., 2021; Chen et al., 2021; Xie et al., 2021b; Zhou et al., 2023), a majority of which follow a hybrid network design, incorporating convolutions in conjunction with Transformer blocks. Notably, recent 3D medical image segmentation architectures using Transformers have increased convolutional components (He et al., 2023b) or utilized Transformers to complement a strong CNN backbone (Chen et al., 2024), as in a MaskFormer (Cheng et al., 2021b;a). This indicates a recent trend of abandoning the Transformer paradigm and moving back in the direction of CNNs for 3D medical image segmentation.

**Why Transformers?** Despite CNNs topping recent large-scale benchmarks (Isensee et al., 2024; Bassi et al., 2024), Transformer architectures remain attractive for medical image segmentation because of the advantages of their sequence modeling paradigm such as: i) **Efficiency in self-supervised learning.** Transformer-based tokenization enables computationally efficient masked image modeling, making large-scale pre-training with unlabeled data cheaper and more scalable. Hence, they are the preferred backbone in modern self-supervised learning, with methods such as Masked Autoencoders (He et al., 2022), I-JEPA (Assran et al., 2023), and DINOv2 (Oquab et al., 2023), DINOv3 (Siméoni et al., 2025), V-JEPA (Bardes et al., 2023) all using Vision Transformers (ViT) (Dosovitskiy et al., 2021). We provide a small example of this efficiency in Table 2; compared to no benefits in a CNN, our proposed architecture uses 40% less VRAM when masking 80% of the tokens. ii) **Multi-modal integration.** Healthcare is inherently multi-modal. Representing high-dimensional 3D images as token sequences simplifies combining visual information with other data sources, e.g., for generating accurate reports from CT or X-rays (Irvin et al., 2019; Johnson et al., 2019; Wang et al., 2022; Stock et al., 2024). This already has strong precedent from natural vision domain with, for example, Chameleon (Team, 2024) integrating text and image tokens together. However, both advantages require a Transformer actually capable of learning strong visual tokens for dense 3D medical image segmentation – a requirement which (we show) most current Transformers do not suitably fulfill and something we directly address. While CNN adaptations of these objectives may sometimes exist, transfer of new paradigms between the natural imaging and medical domain is severely hampered by the fact that both rely on different classes of architectures to achieve state of the art performance.

**Contributions.** In this work, we take a two-pronged approach to establishing an effective Transformer architecture for 3D medical image segmentation.

- Firstly, we examine nine popular Transformer-based architectures for medical image segmentation by quantifying the influence of their Transformer layers. Our analysis reveals that the parameters outside of Transformer blocks in Transformer-CNN hybrids are the primary driver of performance, with the Transformer layers contributing minimally to segmentation performance. In contrast, models with fewer convolution parameters exhibit greater dependence on their Transformer blocks yet consistently underperform compared to pure CNNs. (Section 2)

- Secondly, we revisit Transformer-centric architectures for 3D medical image segmentation and introduce Primus and PrimusV2, the first competitive Transformer-centric architectures. Both architectures minimize convolutional parameters, ensuring that representation learning is driven substantially by Transformer blocks and introduce key innovations such as higher token resolution, adapted 3D rotary positional embeddings, and modern Transformer design elements such as SwiGLU and LayerScale, enabling Primus to achieve competitive performance without relying on convolutions, while PrimusV2 matches ResEnc-L by expanding on Primus through an iterative patch embedding. (Section 3)

## 2 DECONSTRUCTING TRANSFORMERS IN MEDICAL IMAGE SEGMENTATION

In recent years, a handful of Transformer-based architectures (Khan et al., 2023) have established themselves as benchmarks for medical image segmentation, collectively exceeding 24500 citations in the past four years (Table 10). We focus our analysis on nine of the most influential designs – TransFuse (Zhang et al., 2021), TransUNet (Chen et al., 2021), UTNet (Gao et al., 2021), SwinUNet (Cao et al., 2022), SwinUNETR (Hatamizadeh et al., 2021), CoTR (Xie et al., 2021b), nnFormer (Zhou et al., 2023), TransBTS (Wang et al., 2021), and UNETR (Hatamizadeh et al., 2022). These models span both 2D and 3D architectures and are predominantly CNN–Transformer hybrids, summarized in Table 1. A more detailed review of each architecture is provided in Appendix B.3.

### 2.1 HOW MUCH TRANSFORMER IS IN A TRANSFORMER-BASED 3D SEGMENTATION NET?

Current Transformer-based models in 3D medical image segmentation are rarely 'pure' Transformers. Instead, they combine self-attention blocks with substantial non-Transformer components (typically convolutions). To measure this reliance quantitatively, we define the **UNet Index** $= \frac{\text{Model.params} - \text{Model.TR.params}}{\text{UNet.params}}$, where $\text{Model.params}$ is the total parameter count, $\text{Model.TR.params}$ are those inside Transformer blocks, and $\text{UNet.params}$ are parame-

Table 1: **Most segmentation Transformers are CNN-heavy.** Six out of nine architectures have a high UNet index, meaning significant capacity lies outside their Transformer (TR) blocks. In contrast, our Primus and PrimusV2 concentrate the majority of parameters and FLOPs inside the Transformer, enforcing attention usage and realizing a Transformer-centric design.

| Model | Input Dims | Parameters | | | FLOPs | | | UNet Index |
|---|---|---|---|---|---|---|---|---|
| | | Total [M] | In TR [%] | Out TR [%] | Total [B] | In TR [%] | Out TR [%] | |
| TransFuse | 2D | 26.4 | 53.8 | 46.2 | 51.8 | 40.9 | 59.1 | 0.41 |
| TransUNet | 2D | 105.9 | 80.3 | 19.7 | 169.3 | 63.2 | 36.8 | 0.70 |
| UTNet | 2D | 10.0 | 25.6 | 74.4 | 70.8 | 16.3 | 83.7 | 0.25 |
| SwinUNet | 2D | 41.4 | 91.2 | 8.8 | 9.0 | 88.5 | 11.5 | 0.12 |
| SwinUNETR | 3D | 62.2 | 7.9 | 92.1 | 320.9 | 7.9 | 92.1 | 1.91 |
| CoTR | 3D | 41.9 | 22.3 | 77.7 | 281.5 | 13.0 | 87.0 | 1.08 |
| nnFormer | 3D | 37.4 | 62.6 | 37.4 | 26.4 | 45.6 | 54.4 | 0.47 |
| TransBTS | 3D | 31.6 | 66.5 | 33.5 | 120.1 | 40.7 | 59.3 | 0.35 |
| UNETR | 3D | 92.8 | 91.6 | 8.4 | 73.6 | 26.2 | 73.8 | 0.26 |
| Primus-L | 3D | 325.5 | 99.3 | 0.7 | 560.0 | 98.8 | 1.2 | 0.13 |
| PrimusV2-L | 3D | 326.1 | 98.7 | 1.3 | 595.0 | 93.5 | 6.5 | 0.14 |

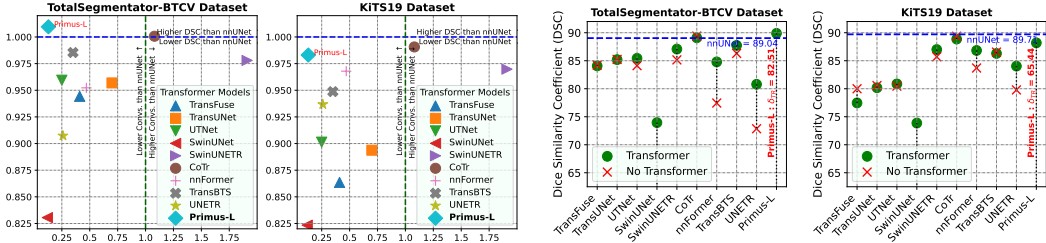

(a) **Transformer Nets vs similarly-trained UNet**  (b) **Transformer Nets vs Nets w/o Transformers**

Figure 1: **No existing nets show low UNet index and high performance.** In Fig. 1a, most architectures fail to beat an *equivalently trained* nnU-Net (8/9 on TotalSegmentator-BTCV, 9/9 on KiTS19). Further, in Fig. 1b, the majority (6/9) lose under 3% performance ($\delta_{TR}$) when Transformers are removed. Finally, Primus is the only low UNet index model competitive with our UNet baseline.

ters of a fixed U-Net model (the nnU-Net of Ulrich et al. (2023) with $\approx$ 30M params). A UNet Index of 1 means the model carries as many convolutional parameters outside its Transformer as the full U-Net, while a value of 2 means twice as many. This is preferable to the `Model.TR.params/Model.params` ratio as it captures cases like TransUNet, which has only 20% convolutions yet a 0.70 UNet-Index.

Across our nine evaluated architectures in Table 1, UNet Indices range from 0.12–0.70 in 2D and 0.26–1.91 in 3D – revealing that most Transformer models actually lean heavily on CNNs. This is, in fact, consistent with their design choices: some simply graft attention blocks onto a pre-existing U-Net backbone (e.g., TransBTS, CoTr, nnFormer), while others rely on convolution-heavy decoders (e.g., UNETR, SwinUNETR). These observations raise a central question: *Are the Transformer blocks truly critical for segmentation, or do they play only an auxiliary role?* To probe this, we conduct controlled experiments on two widely used CT segmentation benchmarks – KiTS19 (Heller et al., 2019) and a 13 BTCV abdominal organ subset of TotalSegmentator (Wasserthal et al., 2022) which we discuss below (with experimental details provided in Appendix A.1.2).

**Experiment 1: Do Transformer-based Nets outperform a well-trained UNet?** We begin by asking the question: Can Transformer-based architectures actually surpass a strong convolutional baseline? To answer this, we train all nine architectures alongside a default nnU-Net (Isensee et al., 2021) under identical settings (same epochs, augmentations, etc.), reporting dice score (DSC) (Zijdenbos et al., 1994) relative to the nnU-Net in Fig. 1a. Surprisingly, nearly all Transformer models fail to outperform the nnU-Net from 2019. This is striking given that several architectures (e.g., CoTr, SwinUNETR) allocate as many parameters outside the Transformer as an entire nnU-Net (UNet Index $\geq$1). The result raises doubts about whether their Transformer blocks, parameter allocation, and overall design meaningfully contribute to segmentation performance. Our model (**Primus**, introduced in Section 3) is the only Transformer-centric architecture competitive with this baseline.

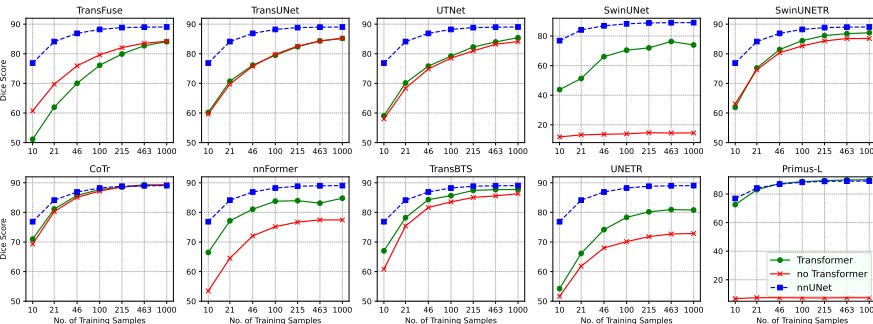

Figure 2: **Scaling Dataset size does not fix issues with existing Transformer-based networks.** Increasing training data on TotalSegmentator-BTCV only increases the gap between **Transformer** and **no Transformer** in 4 out of 9 architectures (UNETR, SwinUNETR, SwinUNet, TransFuse). However, Primus-L effectively uses its Transformer to match the default nnU-Net reference.

**Experiment 2: Stress testing architectures by removing Transformers.** Following the inability to beat a strong UNet, we perform an additional stress test following the premise: *If self-attention is critical, removing the transformer should cause a major drop in performance.* The results (Fig. 1b) show that in many architectures, performance hardly changes at all despite a simple Identity replacing the Transformers, suggesting that most of the representation learning is done or can be done by the CNN backbone alone. In some cases, removing the Transformer even improves accuracy (TransFuse, TransBTS, TransUNet, CoTr), pointing to inefficiencies in how attention is integrated. Only a few models (UNETR, nnFormer, SwinUNet) show a clear reliance on their Transformer blocks – yet these are among the weakest performing models. This highlights that while current architectures integrate Transformers, their contribution to segmentation performance remains limited. We revisit this issue when analyzing learned representations in Appendix E.

## 2.2 Do large Datasets fix this issue?

The weak contribution of Transformers on small medical segmentation datasets invites the question: *Does scaling the dataset size resolve this issue?* The difficulties of training Transformer architectures from scratch on small-scale datasets (Liu et al., 2021a; Touvron et al., 2022; 2021a) are well-known. Therefore, they are commonly pre-trained in the natural image domain (Dosovitskiy et al., 2021). However, in the 3D medical image segmentation domain, there is a lack of such large datasets, with dataset sizes ranging between the low hundreds to (only recently) the high thousands (Wasserthal et al., 2022; Qu et al., 2024). Compounding the problems of low overall size, most datasets used in medical image segmentation are sparsely-labeled to about 20% density while natural image datasets are significantly denser at nearly 90% (Fig. 7 left in Appendix D.2). Therefore, almost all Transformer models are trained from scratch on small, sparsely annotated datasets (Fig. 7 right) (Litjens et al., 2017; Li et al., 2021), potentially leading to the observed difficulties in out-competing state-of-the-art convolutional networks in Section 2.1 and in large-scale benchmarks (Isensee et al., 2024; Bassi et al., 2024).

To test whether limited dataset size explains the weak contribution of Transformers, we study the effect of data scale using the large TotalSegmentator dataset (Wasserthal et al., 2022) using the 13 BTCV abdominal organ labels (Landman et al., 2015). We train all models on subsets scaling between 1–100% of 1000 available samples, and compare them against their versions with Transformers removed. Performance is evaluated on a held-out set of 251 cases. If the performance gap $DSC_{w/TR} - DSC_{wo/TR}$ widens with larger training sets, this suggests that Transformers are limited by data scarcity. Conversely, if the gap remains stable, it implies data scale did not limit the Transformer – as both models naturally improve as data size grows and performance saturates.

In our results presented in Fig. 2, it can be seen that 4 out of 9 architectures (UNETR, SwinUNETR, SwinUNet, TransFuse) perform better with their Transformer blocks included when the data size increases. However, for the remaining 5 out of 9 architectures, no such conclusion can be drawn. Notably, except CoTr at large dataset sizes, no architecture surpasses the U-Net baseline at any scale.

Figure 3: **Primus : A Transformer-centric architecture with minimal convolutions.** Primus extracts high-resolution 3D visual tokens through a strided convolution, while PrimusV2 uses a chain of residual blocks. Once tokenized, both leverage a Transformer featuring 3D axial Rotary Position Embedding (RoPE) and the Eva-02 MLP Block (Fang et al., 2024). A series of Transposed Convolutions forms the decoder, resulting in the final segmentation.

**Conclusion.** Revisiting our results, two consistent patterns emerge – *Firstly*, many architectures rely heavily on non-Transformer parameters, with CNN components alone able to match the performance of the full hybrid model (e.g., CoTr, SwinUNETR, TransBTS, TransUNet, TransFuse). *Secondly*, when models do seem to depend on their Transformer blocks, their overall performance still lags behind a strong U-Net baseline (e.g., nnFormer, UNETR, SwinUNETR, and all 2D architectures).

Our findings also indicate that dataset scale is not the primary bottleneck. Instead, the limitations stem from architectural design choices or insufficiently optimized training strategies. Most existing approaches embed Transformers inside a CNN skeleton, which undermines their potential advantages – such as multi-modal integration and computational efficiency in self-supervised learning. The only partial exception is UNETR, which resembles a vanilla ViT. However, its large performance gap to nnU-Net compromises its practical utility in segmenting 3D medical images.

## 3 PRIMUS: ENFORCING ATTENTION FOR 3D MEDICAL IMAGE SEGMENTATION

Due to the shortcomings of prior architectures, we go back to the drawing board to develop a Transformer architecture for 3D medical image segmentation that is both competitive to state-of-the-art CNNs while staying in the visual token domain by adopting the simplistic elegance of the Vision Transformer architecture paradigm (Dosovitskiy et al., 2021). We revisit prior design decisions made in UNETR, which is the closest Transformer-centric analog (Hatamizadeh et al., 2022), and introduce advancements from the natural language and natural imaging domains to develop a state-of-the-art 3D medical image segmentation Transformer illustrated in Fig. 3. We highlight our improvements in this section, but also mention features which did not influence performance significantly in Appendix C – notable are register tokens (Darcet et al., 2023) and additional attention or projection dropout. Due to our final architecture being the first competitive Transformer-centric architecture for 3D medical image segmentation, we name it **Primus** (*Lat.: "first"*).

### 3.1 ITERATIVE DEVELOPMENT FRAMEWORK

Our development started from a baseline ViT architecture with a fixed minimal tokenizer and a lightweight decoder, to which gradual changes were introduced and rigorously validated on four development datasets, until we converged on our final configuration. To minimize changes being subject to noisy and poorly-labeled radiological data, the recommended benchmarking datasets of Isensee et al. (2024) were used for this, namely: i) *AMOS22* (Ji et al., 2022): Abdominal organ segmentation dataset of 300 CT volumes and 60 MRI volumes with 15 annotated organs. ii) *KiTS23* (Heller et al., 2023): Kidney tumor dataset with 489 CT volumes with annotations provided for kidney, tumor and cysts. iii) *ACDC* (Bernard et al., 2018): 200 cardiac cine-MRI volumes with 3 annotated ventricular structures. iv) *LiTS* (Bilic et al., 2023): Liver tumor segmentation dataset with 131 CT volumes with annotated liver and tumor classes. Due to high computational costs, we used only the first fold (80/20 split) of the default five-fold cross-validation scheme during development, with the remaining folds used in later evaluation experiments (Section 4). The validation results of each incorporated architectural adaptation are highlighted in Table 3.

Table 2: **All Primus configurations.** Only a minority of parameters are bound in convolutions in the tokenizer and decoder. Parameters calculated for $8 \times 8 \times 8$ tokenizer and decoder. UNETR used as transformer reference. *Params: Parameters, TR: Transformer, E.Dim: Embedding Dimension, 50% MAE: VRAM when dropping 50% of tokens*

| Model | Layers | Heads | E.Dim | Primus | | | | PrimusV2 | |
| | | | | Total Pars. | Non-TR Pars. | 50% MAE | 80% MAE | Total Pars. | Non-TR Pars. |
|---|---|---|---|---|---|---|---|---|---|
| **S** | 12 | 6 | 396 | 23.9M | 1.18M | - | - | 24.6M | 1.95M |
| **B** | 12 | 12 | 792 | 93.2M | 2.67M | - | - | 93.8M | 3.29M |
| **M** | 16 | 12 | 864 | 146.6M | 2.96M | - | - | 147.2M | 3.56M |
| **L** | 24 | 16 | 1056 | 325.5M | 3.82M | 10.8Gb | 6.5Gb (-40%) | 326.1M | 4.34M |
| UNETR | - | - | - | 92.8M | 7.8M | - | - | - | - |
| CNN (ResEnc-L) | - | - | - | 102.4M | - | 10.0Gb | 10.0Gb (0%) | - | - |

## 3.2 TOKENIZER & POSITION EMBEDDING

**Rethinking Tokenization for 3D Medical Images.** In order to leverage the Transformer paradigm, the 3D image needs to be converted into a 1D sequence of visual tokens. In Hatamizadeh et al. (2022), this tokenization was conducted through a strided-convolution with kernel size and stride of $16 \times 16 \times 16$, which mirrors the original 2D ViT equivalent (Dosovitskiy et al., 2021). In natural images, this leads to $16 \times 16 \times 3 = 768$ values being encoded into a visual token with identical embedding dimensions, while in 3D medical it represents an immediate compression of dimensionality, potentially removing relevant local information from visual tokens. Especially, in medical image segmentation, this local information is essential to accurately delineate anatomical and pathological structures in radiology images (Zhou et al., 2019; Wei et al., 2023). We introduce two ideas to encode complex fine-grained image features into Transformer tokens as seen in Table 3:

1. **Smaller tokenizer patch sizes:** To maintain local information, we decrease the patch size of the tokenizer to $8 \times 8 \times 8$, allowing tokenization with lower compression. This enables richer representations, at the cost of handling an 8x longer sequence. However, our network design including our lightweight decoder allows this to easily fit within VRAM of current A100 40GB GPUs for even the largest scale of the proposed architectures.

2. **Iterative tokenizer:** Instead of patch-embedding with a single projection, we introduce an iterative patch embedding using convolutional residual blocks (Fig. 3 right). These early convolutions extract low-level features and yield more semantic tokens that the transformer may struggle to learn itself, particularly in limited data settings. Ablations of this are available in Appendix C.2.

**Improving Position Embeddings.** In 3D medical imaging, anatomical structures exhibit strong spatial correlations that are critical for accurate segmentation. While permutation-invariance of the attention mechanism means that Transformers lack inherent spatial understanding, position embeddings are used to overcome this hurdle. However, the sequence shift introduced by patch-based training common to 3D medical image analysis has been seen to decrease positional awareness in absolute positional embedding-based approaches (e.g., sinusoidal or learned) (Sinha et al., 2022). To better capture positional information, we incorporate relative positional embeddings in the form of axial Rotary Position Embeddings (RoPE) (Su et al., 2024), which encode relative distances and orientations directly into the attention mechanism. By extending RoPE to 3D, we ensure that the model is aware of the volumetric nature of medical imaging, capturing nuanced patterns like the relative position of anatomical features and pathological changes. When combined with high-resolution tokens, 3D RoPE allows our model to fully leverage the spatial information present in the data.

## 3.3 ARCHITECTURE AND TRAINING IMPROVEMENTS

**SwiGLU MLP Blocks.** Following advances in NLP and computer vision in natural images, Gated Linear Units (GLUs) have succeeded default ViT MLP blocks (Shazeer, 2020). Recently, Swish-Activation GLUs (SwiGLU), as introduced in Fang et al. (2024) with an additional LayerNorm block, showed state-of-the-art performance in the natural imaging domain. We found this to translate to dense 3D medical image segmentation and hence adopted this MLP block structure.

**Lightweight low-convolutional decoder.** To limit the number of convolutional layers, we use a lightweight decoder composed of a sequence of back-to-back transposed convolution, normalization, and activation (`TPConv-Norm-Act`) blocks projecting the tokens back into full-resolution image

Table 3: **Ablation of the introduced changes to a default ViT architecture.** We consecutively decreased patch size, added a 3D rotary positional embedding (RoPE), the EVA02-MLP head and Drop Path. Colored rows indicate the same configuration across tables. *LPe: Learnable Positional Embedding, DP: Drop Path, LS: LayerScale, PAN: Post Attention Normalization, LR: Learning Rate, PE+: Iterative Patch Embedding *: Unstable runs that were repeated with lower LR (3e-5).*

| | | Configurations | | | | | | Dice Similarity Coefficient (in %) | | | | |
|---|---|---|---|---|---|---|---|---|---|---|---|---|
| MLP-Block | Token PS | LPe | 3D RoPE | DP | LS | PAN | PE+ | ACDC | AMOS22 | KiTS23 | LiTS | Avg. |
| ViT | [16x16x16] | ✓ | ✗ | ✗ | ✗ | ✗ | ✗ | 90.74 | 78.84 | 66.78 | 71.68 | 77.01 |
| ViT | [8x8x8] | ✓ | ✗ | ✗ | ✗ | ✗ | ✗ | 91.70 | 79.81 | 70.74* | 75.18* | 79.36 |
| ViT | [8x8x8] | ✓ | ✓ | ✗ | ✗ | ✗ | ✗ | 92.36 | 87.55 | 86.90 | 82.53 | 87.34 |
| ViT | [8x8x8] | ✓ | ✓ | ✓ | ✗ | ✗ | ✗ | 92.64 | 87.89 | 87.48 | 81.58 | 87.40 |
| EVA02 | [8x8x8] | ✓ | ✓ | ✗ | ✗ | ✗ | ✗ | 92.48 | 87.98 | 87.36 | 82.36 | 87.55 |
| EVA02 | [8x8x8] | ✓ | ✓ | ✓ | ✗ | ✗ | ✗ | 92.73 | 87.89 | 88.11 | 82.89 | 87.91 |
| EVA02 | [8x8x8] | ✓ | ✓ | ✓ | ✓ | ✓ | ✗ | 92.74 | 88.48 | 87.74 | 82.42 | 87.85 |
| EVA02 | [8x8x8] | ✓ | ✓ | ✓ | ✓ | ✓ | ✓ | **92.81** | **89.34** | **88.57** | **84.91** | **88.91** |
| **Configurations with Transformer replaced by Identity** | | | | | | | | | | | | |
| EVA02 | [8x8x8] | ✓ | ✓ | ✓ | ✓ | ✓ | ✗ | 19.45 | 03.02 | 15.88 | 40.76 | 19.78 |
| EVA02 | [8x8x8] | ✓ | ✓ | ✓ | ✓ | ✓ | ✓ | 91.89 | 81.56 | 82.69 | 79.77 | 83.98 |

space. This minimizes UNet-index and consequently convolutional influence, and simultaneously frees up VRAM to process the longer visual token sequence.

**Improved Stability and Regularization.** Our large Primus-L architecture encountered stability issues while training on KiTS23 and LiTS. While lower learning rates can solve this problem, it also reduces performance and requires manual intervention. We mitigate this by using LayerScale (Touvron et al., 2021b) which introduces additional learnable parameters applied to the output of Attention and MLP blocks. Additionally, a post-attention Layer Normalization stabilizes our network further, with some datasets even showing an overall increase in performance due to this adaptation, as visualized in Table 5. Finally, DropPath (Larsson et al., 2016) and high degrees of weight decay were found to positively influence Transformer performance (Table 3).

**Scaling the Network.** Following our changes, we introduce four configurations of differing scales – Primus(V2)-S/B/M/L, which are fully defined by their number of layers, heads, and embedding depth as visualized in Table 2. As highlighted, Primus and PrimusV2 feature low UNet-index with a minimal amount of convolution parameters, while allocating most parameters to the Transformer.

Table 4: 3D RoPE improves performance and stabilizes training. Results of Primus-M. *LPe: Learned Position Embedding; Red row indicates identical configuration of Table 3.*

| Pos. embedding | | Dice Similarity Coefficient | | | | |
|---|---|---|---|---|---|---|
| LPe. | 3D RoPE | ACDC | AMOS22 | KiTS23 | LiTS | Avg. |
| ✗ | ✗ | 61.89 | 49.48 | 38.61 | 62.51 | 53.12 |
| ✓ | ✗ | 91.45 | 84.22 | 59.16 | 73.69 | 77.13 |
| ✗ | ✓ | 92.36 | 87.75 | 87.79 | 83.46 | 87.84 |
| ✓ | ✓ | 92.73 | 87.89 | 88.11 | 82.89 | 87.91 |
| | | *Lower LR (3e-5)* | | | | |
| ✗ | ✗ | - | - | 48.93 | 64.48 | - |
| ✓ | ✗ | - | - | 73.10 | 75.80 | - |
| ✗ | ✓ | - | - | 87.05 | 81.27 | - |

Table 5: Layer Scale and Post Attention Normalization improves convergence stability. *LS: Layer Scale, PAN: Post Attention Normalization, LR: Learning Rate.*

| Config. (LR) | LS | PAN | ACDC | AMOS22 | KiTS23 | LiTS | Avg. |
|---|---|---|---|---|---|---|---|
| S | ✗ | ✗ | 92.36 | 87.15 | **87.10** | 82.51 | 87.28 |
| Primus-S | ✓ | ✓ | **92.46** | **87.47** | 86.76 | **82.89** | **87.40** |
| B | ✗ | ✗ | 92.63 | 87.76 | **88.03** | 83.03 | **87.86** |
| Primus-B | ✓ | ✓ | **92.70** | **87.87** | 86.83 | **83.16** | 87.64 |
| M | ✗ | ✗ | 92.73 | 87.89 | **88.11** | **82.89** | **87.91** |
| Primus-M | ✓ | ✓ | **92.74** | **88.48** | 87.74 | 82.42 | 87.85 |
| L (3e-4) | ✗ | ✗ | 92.41 | 88.24 | 86.89 | 67.47 | 83.75 |
| L (3e-5) | ✗ | ✗ | 92.69 | 88.28 | 87.39 | 81.81 | 87.54 |
| Primus-L | ✓ | ✓ | **92.71** | **88.60** | **88.64** | **83.00** | **88.24** |

## 4 EXPERIMENTS

All experiments were conducted in the nnU-Net framework (Isensee et al., 2021) and all architectures were trained for 1000 epochs with 250 steps per epoch. All runs of Primus are trained with a learning rate of 3e-4, weight decay of 5e-2, AdamW optimizer, and gradient clipping of 1 unless otherwise noted. The drop ratio of DropPath was set to 0.2 and Layer Scale was initialized with 0.1. Data was preprocessed through nnU-Net's automatic preprocessing for all datasets but ACDC, for which a 1mm isotropic spacing was chosen, as proposed in Isensee et al. (2024). No interventions had to be taken to adapt the planned input patch size of the CNN to Primus. The *'L'* configurations were only trained for the folds of the development dataset due to computational constraints.

As baselines, we compare against i) default nnU-Net (Isensee et al., 2021), ii) Residual Encoder L

Table 6: **Test results.** Average DSC of the development dataset folds 1 to 4, and all five folds of the test datasets. Colored rows indicate the same configuration across tables. We highlight the top-3 performance using **best**, second-best and third-best per column.

| Architectures | Development Datasets | | | | Test Datasets | | | | | AVG |
|---|---|---|---|---|---|---|---|---|---|---|
| | ACDC | AMOS22 | KiTS23 | LiTS | SST3 | MAMA | SBM | Atlas22 | Word | |
| **Convolutional Baselines** | | | | | | | | | | |
| nnUNet def. | 91.34 | 88.61 | 85.99 | 79.29 | 90.27 | 78.32 | **66.52** | 63.11 | 83.11 | 80.73 |
| nnUNet ResEnc-L | 92.54 | 89.39 | 88.06 | 81.20 | **90.28** | 79.00 | 64.00 | 63.12 | **85.79** | 81.48 |
| MedNeXt-L | 92.55 | **89.58** | **88.20** | 81.57 | 89.93 | 79.42 | 65.85 | 63.03 | 85.37 | **81.72** |
| **Other Baselines** | | | | | | | | | | |
| SegMamba | **92.65** | 86.60 | 82.52 | 76.89 | 89.19 | 75.67 | 63.58 | 62.23 | 82.95 | 79.14 |
| VisionxLSTM | 92.41 | 78.30 | 80.49 | 74.61 | 88.32 | 73.54 | 61.04 | 60.09 | 74.85 | 75.96 |
| **Transformer Baselines** | | | | | | | | | | |
| CoTR | 90.50 | 87.93 | 84.63 | 78.44 | 89.60 | 76.95 | 59.96 | 62.14 | 83.11 | 79.25 |
| nnFormer | 92.35 | 81.35 | 75.72 | 77.02 | 88.62 | 68.71 | 64.37 | 60.61 | 82.53 | 76.81 |
| SwinUNETR | 91.11 | 80.88 | 75.07 | 73.27 | 88.60 | 75.55 | 61.96 | 60.56 | 78.99 | 76.22 |
| UNETR | 90.41 | 62.93 | 76.33 | 70.91 | 86.73 | 74.27 | 49.87 | 54.15 | 70.87 | 70.72 |
| UNETR++ | 92.58 | 84.82 | 82.42 | 77.69 | 89.50 | 77.16 | 64.33 | 61.39 | 77.72 | 78.62 |
| SwinUNETR-v2 | 91.94 | 86.17 | 83.99 | 77.18 | 88.72 | 75.84 | 60.80 | 61.10 | 82.20 | 78.66 |
| **Primus-V1** | | | | | | | | | | |
| Primus-S | 91.82 | 87.47 | 85.87 | 78.97 | 87.83 | 77.14 | 56.71 | 60.21 | 82.84 | 78.76 |
| Primus-B | 92.07 | 88.08 | 86.37 | 79.26 | 88.33 | 76.17 | 57.62 | 60.80 | 83.01 | 79.08 |
| Primus-M | 92.26 | 88.18 | 86.38 | 79.52 | 88.31 | 76.39 | 57.63 | 60.10 | 82.98 | 79.08 |
| Primus-L | 92.10 | 88.69 | 86.67 | 79.90 | - | - | - | - | - | - |
| **Primus-V2** | | | | | | | | | | |
| PrimusV2-S | 92.37 | 88.95 | 87.68 | 80.61 | 88.69 | **79.55** | 66.22 | 62.98 | 83.98 | 81.23 |
| PrimusV2-B | 92.35 | 89.31 | 88.14 | 80.97 | 88.48 | 79.20 | 65.24 | 63.14 | 83.83 | 81.18 |
| PrimusV2-M | 92.27 | 89.35 | 88.09 | **81.73** | 88.26 | 79.40 | 66.36 | **63.23** | 84.15 | 81.43 |
| PrimusV2-L | 92.36 | 89.27 | 88.07 | 81.28 | - | - | - | - | - | - |

U-Net (ResEnc-L) (Isensee et al., 2024), iii) nnFormer (Zhou et al., 2023), iv) CoTR (Xie et al., 2021b), v) UNETR (Hatamizadeh et al., 2022) and vi) SwinUNETR (Hatamizadeh et al., 2021), vii) MedNeXt (Roy et al., 2023), viii) UNETR++ (Shaker et al., 2024), ix) SwinUNETRv2 (He et al., 2023b), x) SegMamba (Xing et al., 2024), xi) VisionxLSTM (Dutta et al., 2025) as they represent strong CNN, Transformer baselines or alternative state-of-the-art methodologies such as Mamba (Gu & Dao, 2024) or xLSTM (Beck et al., 2024). An exception is UNETR, which is a weaker baseline but represents the most similar architecture to Primus. We provide the dataset-specific, detailed hyperparameters used for training in Appendix A.2 and Table 8.

**Test Datasets** After the development of Primus, we trained the final configurations and the baseline models for the remaining four folds on the development datasets and extended the dataset collection by five additional, previously untouched test datasets. The test datasets chosen are: i) *StructSeg Task 3 (SST3)*: 50 head & neck CT volumes with annotations for 5 organs-at-risk in radiotherapy (Li et al., 2019). ii) *MAMA MIA (MAMA)* (Garrucho et al., 2024): Breast cancer dataset of 1506 volumes of dynamic contrast-enhanced magnetic resonance images (DCE-MRI) with annotated tumor segmentations. iii) *Stanford Brain Metastases (SBM)* (Grøvik et al., 2020): 105 Whole Brain MRI volumes with cerebral metastasis annotations of at least one per scan. iv) *Atlas22* (Liew et al., 2022): The R2.0 variant of the dataset contains 655 T1-weighted (T1w) MRI brain volumes with annotated stroke lesions. v) *WORD* (Luo et al., 2022): A dataset of 120 abdominal CT volumes with 16 annotated organs. These datasets were chosen as they represent a diverse set of modalities, body-regions, disease characteristics and segmentation structures.

## 5 RESULTS AND DISCUSSION

**State-of-the-art Transformer-based Networks.** We report mean 5-fold cross-validation results on the final Primus configurations across all development and test datasets, excluding the first fold of the development sets used for hyperparameter tuning, in Table 6, with qualitative results visualized in Fig. 4. Across datasets, our Primus configuration outperforms other hybrid transformers and is competitive with the default nnU-Net, exceeding it on four out of nine datasets. However,

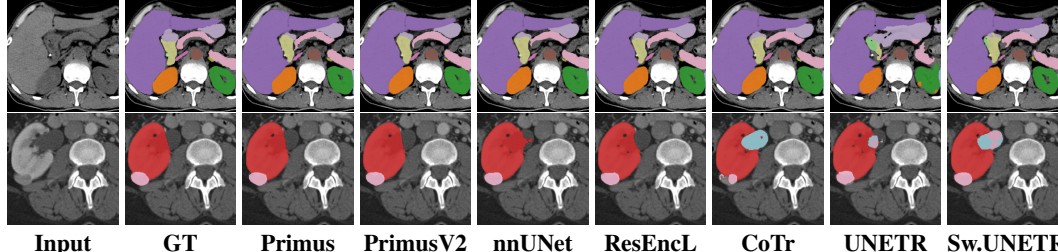

| Input | GT | Primus | PrimusV2 | nnUNet | ResEncL | CoTr | UNETR | Sw.UNETR |

Figure 4: **Qualitative Results.** Primus V2 is on par with SOTA CNNs, while baselines lag behind. *Top row: AMOS22 dataset; Bottom row: KiTS23 dataset*

PrimusV2 takes this further, outperforming nnUNet on 7 out of 9 datasets and reaching parity with ResEnc-L. Across newer baselines such as MedNeXt-L, PrimusV2 exceeds it on 4 out of 9 cases. This is significantly more than *any* of the six Transformer baselines evaluated by us including state-of-the-art ones such as UNETR++ and SwinUNETRv2. We also outperform state-space (SegMamba) and xLSTM models (VisionxLSTM) by decent margins.

PrimusV2 also exceeds all popular Transformer baselines such as CoTr, nnFormer, and Swin-UNETR, while allocating fewer parameters to non-Transformer components. Compared to the closest Transformer-centric counterpart UNETR, all Primus configurations exceed it by an average of 8 and PrimusV2 by 11 DSC points across all datasets. This outperformance also extends to newer baselines such as SwinUNETRv2 by 2.77 DSC. More importantly, PrimusV2 is the ***only*** Transformer-based architecture to be within 0.05 DSC and 0.29 DSC of state-of-the-art CNNs such as ResEnc-L and MedNeXt respectively. Infact, ResEnc-L, MedNeXt-L and PrimusV2 are almost always (7 out of 9 cases) among the top three networks in our benchmarking across our datasets. This demonstrates that our Transformer-centric designs are viable even in small and sparsely annotated medical data regimes. In doing so, we note that this marks a fundamental paradigm shift, as CNNs with their inductive biases are no longer the *only* reliable choice for 3D medical image segmentation.

**Iterative Tokenizer improves Encoding.** While the smaller tokenizer patch size in Primus already improves performance, our usage of an iterative Patch Embedding, as in PrimusV2, allows substantial gains of 2 DSC points across all datasets. In particular for SBM, the brain metastases dataset with small brain metastases lesions, which are commonly $< 0.05cm^3$ in median volume (Pflüger et al., 2022), performance increases about 10 DSC points. We believe this to be due to the patch embedding failing to identify small, relevant hyperintense regions with its $8 \times 8 \times 8$ kernel, which we discuss in depth in Appendix C.3. Overall, PrimusV2 exceeds ResEnc-L on 5 out of 9 datasets evaluated, while being within 0.1 DSC of the average of ResEnc-L across datasets. This highlights the importance of the tokenization strategy, which respects the challenges of the domain, such as capturing small granular structures in 3D medical images, for which large strided $16 \times 16 \times 16$ Convolutions of existing designs are not suited.

**Model Scaling vs. Small Medical datasets.** Focusing on the performance between Primus configurations, it can be observed that larger architecture scales do not improve performance consistently, with scaling being heavily dataset-dependent. We observe performance increases when scaling our architecture on AMOS22 and LiTS up to our M configuration – a finding consistent with the observed scaling behavior of Isensee et al. (2024) on medical datasets. This scaling stalls for the L configuration, which we hypothesize may indicate it is not trained for long enough to reach convergence. Regarding very small datasets like SST3, having only 50 total samples (40 train, 10 validation during 5-fold cross-validation), increasing the architecture scale reduces performance. This is not surprising when training a 100M parameter model from scratch, given the severely limited training sample size.

**CNN-Transformer Hybrid Architectures.** While our analysis on Transformer under-performance focuses on a number of existing hybrid architectures, we do not consider this an indictment on hybrid CNN-Transformer architectures themselves. Rather, our focus is on the effectiveness of the Transformer itself in driving segmentation performance. In fact, PrimusV2 demonstrates a ∼5.0 DSC degradation on removing the Transformer while it is much larger at ∼68.0 DSC without the

Transformer in PrimusV1 with a smaller tokenizer (Table 3). This indicates that PrimusV2 is much more of a *hybrid* architecture with the tokenizer more adept at learning coarser features to maintain a higher baseline performance, while simultaneously having a stronger Transformer under-the-hood. We contrast this against architectures such as TransUNet or CoTr which show near 0.0 DSC of degradation without a Transformer (Fig. 1b), thereby having CNNs obscure this underperformance. Therefore, the effectiveness of a powerful Transformer backbone and a strong CNN tokenizer in PrimusV2 demonstrates that it is a stronger combination than either alone.

# 6 LIMITATIONS AND CONCLUSION

In this work, we examined established hybrid Transformer architectures for 3D medical image segmentation and found that most either (i) rely heavily on non-Transformer parameters or (ii) fall short of a simple U-Net baseline. Moreover, those that do perform well typically achieve similar results without meaningful reliance on their Transformer blocks, and sacrifice compatibility with modern Masked Image Modeling (MIM) strategies for efficient pre-training.

To address this, we introduce the Primus and PrimusV2 architecture families, both Transformer-centric networks customized to the unique challenges in 3D medical image segmentation. Primus makes substantial progress in closing the gap to the default nnU-Net, exceeding it on many datasets while PrimusV2 expands on this through a larger, iterative patch embedding to reach parity with the strong ResEnc-L CNN baseline (Isensee et al., 2024).

Nevertheless, these designs are not without limitations. Primus shows reduced performance, in particular on small structures, which diminishes its overall utility. Enlarging the patch embedding in PrimusV2 mitigates this limitation but introduces its own: i) We acknowledge that the iterative tokenizer leverages residual blocks, which increase UNet index and therefore, overall convolutional dependency. While this may seem detrimental to the contributions of the Transformer, we still found PrimusV2 to depend heavily on it to perform well. Moreover, its improved performance, in particular on small lesions, we deem this to be an acceptable compromise. ii) The current patch embedding leads to an overlapping field-of-view of the tokens, deviating from the original Vision Transformer paradigm. This undesired side effect may influence the efficacy of MIM pre-training paradigms, particularly when using random masking. While this may seem problematic at first, a larger masking ratio or adopting block masks can most likely alleviate this.

Despite these limitations, we believe Primus and PrimusV2 represent the first strong Transformer-centric architectures that produce meaningful visual tokens in 3D medical image segmentation, opening up future possibilities to be seamlessly integrated into multi-modal pipelines and to be effectively leveraged in self-supervised learning. Moreover, with suitable supervised or self-supervised pre-training, we expect these architectures to not only match but to surpass the established CNN baselines such as ResEnc-L, thereby enabling state-of-the-art 3D medical image Transformers.

# 7    REPRODUCIBILITY STATEMENT

Our proposed architecture Primus and PrimusV2 was developed in the popular nnU-Net framework and leverages its dynamic planning and preprocessing. The majority of the used datasets are well-established in the domain and are accessible through *'Grand-Challenge'*, *'Zenodo'*, or other public hosts. Combined, this allows for easy reproduction of the preprocessed data as well as the five-fold cross-validation splits. Moreover, as we will share dedicated nnU-Net trainers which use our architecture, we empower all nnU-Net users to reproduce our Primus experiments, making it widely reproducible.

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

Figure 5: **Segmentation performance pre-and-post Identity replacement of a Transformer module quantifies their importance.** By replacing the entire Transformer block, including LayerNorm, Multi-Head Self-Attention or Shifted Window Multi-head Self-Attention, the influence of the entire Transformer within an architecture can be evaluated.

## A EXPERIMENT DETAILS

In the following sections, we provide details on the experiments highlighted in the main paper. In Appendix A.1 we provide details on how the replacement was conducted and how the hyperparameters of the training were configured. The training details for each dataset of the Primus method development and final tests, are provided in Appendix A.2

### A.1 TRANSFORMER INTROSPECTION DETAILS

#### A.1.1 TRANSFORMER REPLACEMENT DETAILS

In order to measure the influence of an introduced Transformer block in the hybrid architectures, we replace the respective blocks with an identity mapping, which simply forwards the input directly to the output, see Fig. 5. For both, the original architectures and the identity-replaced architecture, we train three random seeds on the AMOS-CT and KiTS19 dataset to get robust results. While this replacement is simple for the case of self-attention blocks, UTNet (Gao et al., 2021) employs cross-attention during upsampling. The cross-attention is computed between the representations of the encoder – passed by skip connections – with the lower resolution representations. As we do not want to cut off either of the streams, a simple identity replacement is not possible. Instead, we introduce a 1x1x1 Convolution to project the lower-resolution stream to the same channel dimension as the skip connection stream (compressing it) and then utilize a bilinear interpolation to upsample it to the same spatial resolution as the stream passed by the skip connection. Both streams of identical dimensionality are then added, and processing proceeds normally. This allows maintaining the overall structure of the architecture, while removing any attention mechanisms and introducing minimal additional parameters.

#### A.1.2 TRANSFORMER INTROSPECTION TRAINING CONFIGURATIONS

For all architectures, three randomly initialized seeds were trained to improve stability of results. The network training scheme of Section 2.1 is based heavily on the nnU-Net default settings of the nnUNet v1 framework with minor changes added on top of the nnUNet framework (Isensee et al., 2021). To maintain comparability between all architectures, their input patch size was fixed to $[96 \times 96 \times 96]$ for all 3D networks and $512 \times 512$ for all 2D networks (except SwinUNet whose configuration of the Swin Transformer architecture required an input patch size of $224 \times 224$). The AdamW optimizer (Loshchilov & Hutter, 2017) was used as the optimizer with 1e-4 as the learning rate for all networks incorporating a ViT (namely TransFuse, TransUNet, UTNet, TransBTS and UNETR) and 5e-4 as that of all Swin-based networks (namely, SwinUNet and SwinUNETR ). An exception is SwinUNet which showed unstable training performance with 5e-4 and thus needed a lower learning rate of 1e-4. Table 7 provides a detailed description of the training settings. For nnFormer and CoTR, which were proposed within the nnU-Net framework, we used their recommended SGD optimizer with learning rate of 1e-2.

While we train for 1000 epochs in all experiments, in the experiments conducted on the KiTS19 and AMOS-CT dataset each epoch comprises 250 steps – the default of nnU-Net. For the experiments on dataset size scaling (conducted on the TotalSegmentator-BTCV dataset), we increased the total amount of steps from 250 to 500 steps per epoch. This was done to guarantee that architec-

tures reached their saturation performance, as we expect the larger amount of training samples – maximally 1000 – to require more iterations to converge to the final solution.

We maintain consistency in hyperparameters whether we are using these architectures in Section 2.1 or when conducting the dataset scaling experiment in Section 2.2.

**Datasets**  We train and evaluate the architectures with and without the Transformer blocks on three datasets in our first experiments. AMOS-CT (Ji et al., 2022), KiTS19 (Heller et al., 2019) and a variant of TotalSegmentator, where we only train on the classes of TotalSegmentator (Wasserthal et al., 2022) that correspond to the BTCV dataset (Landman et al., 2015). In the following, we provide some more details for each of these datasets:

1. *TotalSegmentator-BTCV*: The TotalSegmentator dataset is one of the earliest large-scale CT datasets with 117 structures annotated in more than 1000 CT images. The large number of classes makes this dataset unwieldy for large-scale ablation experiments; hence, we sought a compromise by using the 13 classes in the massively popular Beyond-The-Cranial-Vault (BTCV) dataset. We filtered the dataset for CT volumes which contained all 13 classes, which allowed 1251 CT volumes for our dataset scaling experiments.

2. *KiTS19*: The Kidney Tumor Segmentation (KiTS) 2019 challenge was organized for the development of techniques for the segmentation of 2 classes - kidney and tumors in abdominal CT scans. The 210 CT volumes of this dataset used in this study are publicly available.

3. *AMOS-CT*: The Multi-Modality Abdominal Multi-Organ Segmentation (AMOS) Challenge 2022 was a public competition for the automated segmentation of 15 abdominal organs in CT images. The organizers subsequently released 300 CT images and their corresponding segmentation masks post-competition for public usage, which is used by us in this work.

Table 7: The training details of all networks are provided. The hyperparameters are constant for training during all experimental modes - low dataset experiments or network modification experiments. Some hyperparameters are the default settings[†] of the nnUNet framework.

| Architecture | Learning Rate | Weight Decay | Optimizer | Data Augmentation | Patch Size | Authors used nnUNet |
|---|---|---|---|---|---|---|
| SwinUNet[2D] (Cao et al., 2022) | $5e-4$ | | | | $224 \times 224$ | × |
| TransFuse[2D] Zhang et al. (2021) | | | | | | × |
| TransUNet[2D] Chen et al. (2021) | $1e-4$ | | | | $512 \times 512$ | × |
| UTNet[2D] Gao et al. (2021) | | | | | | × |
| SwinUNETR[3D] Hatamizadeh et al. (2021) | $5e-4$ | $3e-5$[†] | AdamW | nnUNet Default[†] | | × |
| TransBTS[3D] Wang et al. (2021) | $1e-4$ | | | | | × |
| UNETR[3D] Hatamizadeh et al. (2022) | | | | | $96 \times 96 \times 96$ | × |
| CoTr[3D] Xie et al. (2021b) | | | | | | ✓ |
| nnFormer[3D] Zhou et al. (2023) | $1e-2$[†] | | SGD[†] | | | ✓ |
| nnUNet[3D] Isensee et al. (2021) | | | | | | – |
| nnUNet[2D] Isensee et al. (2021) | | | | | $512 \times 512$ | – |

## A.2  DEVELOPMENT AND TEST DATASET CONFIGURATIONS

To compare our Primus architecture against the baselines, we implemented all baselines as well as Primus within the nnU-Net framework (Isensee et al., 2021), which uses PyTorch. As nnU-Net conducts unique planning and preprocessing for each dataset based on a dataset fingerprint, we list the preprocessing details for each dataset used in the study in Table 8. All methods are trained for an equal amount of 1000 epochs with each epoch comprising 250 steps. For each method, we provide further training details on their configurations in the following paragraph. Keep in mind that input patch size, batch size, and spacing are defined by the specific plans, each architecture follows, if not specified otherwise:

1. **nnU-Net Default** (Isensee et al., 2021): Following the default nnU-Net v2 configurations, namely Learning Rate 1e-2, Weight Decay 3e-5, gradient clipping 12, optimizer SGD with Nesterov and momentum 0.99 and the default nnU-Net PolyLR Scheduler

2. **nnU-Net ResEnc-L** (Isensee et al., 2024): Hyperparameters are the same as for nnU-Net Default, however, the encoder architecture changes to a residual encoder U-Net with various number of residual blocks per stage, as highlighted in Table 8.

3. **nnFormer** (Zhou et al., 2023): nnFormer is not a singular architecture, but the authors in fact propose three different architecture configurations for ACDC, the BTCV dataset (Landman et al.,

2015) and one for brain tumor segmentation (BraTS). Due to the vastly different parameterizations, we follow Isensee et al. (2024) and use their reported results. In their paper, they adapted nnFormer to the closest configuration the authors proposed for ACDC, AMOS22, KiTS22, and LiTS. For the remaining datasets in our test suite, we chose the *nnFormer tumor* configuration, which is the smallest configuration and likely the best configuration given the dataset training sample sizes of the remaining test datasets. As nnFormer was proposed to be used with nnU-Net (Isensee et al., 2021) plans, we use the default nnU-Net plan pre-processed data for it. The Hyperparameters were identical to the nnU-Net defaults, as per their paper.

4. **CoTr** (Xie et al., 2021b): Opposed to the prior architectures CoTr is a singular, fixed architecture, that – similar to nnFormer – leverages the nnU-Net framework in their repository[1]. Subsequently, we follow the nnU-Net default plans for CoTr. Moreover, to ensure compatibility with small datasets, we round each dimension to the next closest divisor of 16, as is required by the architecture. This change affects the ACDC and the MAMA MIA dataset, slightly increasing the input patch size. Regarding hyperparameters, we follow the ones proposed in the repository, which are identical to the nnU-Net default as well.

5. **SwinUNETR**(Hatamizadeh et al., 2021): In contrast to prior methods, SwinUNETR was proposed and developed in the MONAI framework and, hence was not proposed with nnU-Net's dynamic planning strategy. As they do not specify a set spacing though, we choose to follow the nnU-Net default planning strategy, but due to architectural and VRAM constraints set its patch size to a fixed value of $[96 \times 96 \times 96]$ and the batch size to a fixed value of 2. Other hyperparameters were chosen to: Learning Rate 5e-4, Weight Decay 3e-5, gradient clipping 12, optimizer AdamW with eps 1e-4, as well as a PolyLR learning rate schedule. In this configuration, the learning rate was reduced from the originally proposed learning rate due to convergence stability problems .

6. **UNETR** (Hatamizadeh et al., 2022): Analogously to SwinUNETR, UNETR proposed a constant patch size of $[96 \times 96 \times 96]$ which we adopted and employed for all our experiments. Moreover, in the original paper, the authors propose to resample to $[1 \times 1 \times 1]$ spacing, which we do not follow. Instead, we follow the ResEnc-L plan to remove the effect of different spacing choices. We only do this intervention for UNETR, because it is the closets pure Transformer architecture to Primus and because no explicit information was given on how it should be trained. Aside from this, the hyperparameters chosen were Learning Rate 1e-4, weight decay 3e-5, gradient clipping 12, and AdamW optimizer with eps 1e-4. Due to the original description in the paper (Hatamizadeh et al., 2022) not referencing any learning rate scheduling, we train this with a static learning rate and no scheduling.

7. **Primus**: Primus is integrated into the nnU-Net framework, hence we follow the nnU-Net planning strategy. However, we follow the more recent nnU-Net ResEnc-L planning strategy, resulting in larger input patch sizes and lower batch sizes. While all our Primus experiments in this paper follow this strategy, Primus is not limited to this planning strategy and could be used in conjunction with other plans. Hyperparameters are: Learning Rate 3e-4, Weight decay 5e-2, gradient clipping 1, Drop Path 0.2, Layer Scale 0.1, optimizer AdamW with eps 1e-8 and betas (0.9, 0.98) and fused set to 'True'. While we provide these values, we recommend checking out the repository, which holds the official implementations of the Primus trainers to allow reproduction.

## B  RELATED WORK

The successes of the Transformers in classification, segmentation and detection (Khan et al., 2022; Liu et al., 2023; Amjoud & Amrouch, 2023; Li et al., 2023) tasks in natural images drove their adaptation in deep neural networks for medical image segmentation. However, while the massive amounts of data in the natural image domain drove Transformer-based segmentation techniques with limited inductive bias (Zheng et al., 2021; Strudel et al., 2021; Xie et al., 2021a), research in medical image segmentation steered towards pairing Transformers with convolutional networks with relatively higher inductive biases for effectively learning representations for medical image segmentation. Vision Transformers (Dosovitskiy et al., 2021) enable the learning of long-range global dependencies in visual domains. Swin-Transformers (Liu et al., 2021b), on the other hand, use local shifted-window attention to enable local representation learning. Both use strided non-overlapping

---

[1]https://github.com/YtongXie/CoTr/tree/main

Table 8: **Dataset and architecture configuration details.** nnU-Net conducts unique preprocessing and architecture planning based on the dataset fingerprint and the targeted VRAM. We provide the associated plans for nnU-Net default and ResEnc-L as used in Isensee et al. (2024). While input patch size, batch size and architecture vary for different plans the spacing is identical, with the exception of the ACDC dataset, where nnU-Net ResEnc-L was proposed with an isotropic spacing (Isensee et al., 2024). Our Primus configurations follow the ResEnc-L plan. *Z-score: Z-Score normalization, CT: CT Normalization, BS: Batch Size, IPS: Input Patch Size.*

| Dataset | Plans | Normalization | Spacing | BS | IPS | nnU-Net Configuration | |
| | | | | | | Downsampling strides | Convs/Blocks per stage |
|---|---|---|---|---|---|---|---|
| ACDC | Default | Z-Score | [5.0, 1.56, 1.56] | 4 | [20, 256, 224] | [[1, 1, 1], [1, 2, 2], [2, 2, 2], [2, 2, 2], [1, 2, 2], [1, 2, 2]] | [2, 2, 2, 2, 2, 2] |
| ACDC | ResEnc-L | Z-Score | [1.0, 1.0, 1.0] | 3 | [96, 256, 256] | [[1, 1, 1], [1, 2, 2], [2, 2, 2], [2, 2, 2], [2, 2, 2], [1, 2, 2], [1, 2, 2]] | [1, 3, 4, 6, 6, 6, 6] |
| AMOS22 | Default | Z-Score | [2.0, 0.71, 0.71] | 2 | [64, 160, 192] | [[1, 1, 1], [1, 2, 2], [2, 2, 2], [2, 2, 2], [2, 2, 2], [2, 2, 2]] | [2, 2, 2, 2, 2, 2] |
| AMOS22 | ResEnc-L | Z-Score | [2.0, 0.71, 0.71] | 2 | [96, 224, 224] | [[1, 1, 1], [1, 2, 2], [2, 2, 2], [2, 2, 2], [2, 2, 2], [2, 2, 2]] | [1, 3, 4, 6, 6, 6] |
| KiTS23 | Default | CT | [1.0, 0.78, 0.78] | 2 | [128, 128, 128] | [[1, 1, 1], [2, 2, 2], [2, 2, 2], [2, 2, 2], [2, 2, 2], [2, 2, 2]] | [2, 2, 2, 2, 2, 2] |
| KiTS23 | ResEnc-L | CT | [1.0, 0.78, 0.78] | 2 | [160, 224, 192] | [[1, 1, 1], [2, 2, 2], [2, 2, 2], [2, 2, 2], [2, 2, 2], [2, 2, 2]] | [1, 3, 4, 6, 6, 6] |
| LiTS | Default | CT | [1.0, 0.77, 0.77] | 2 | [128, 128, 128] | [[1, 1, 1], [2, 2, 2], [2, 2, 2], [2, 2, 2], [2, 2, 2], [2, 2, 2]] | [2, 2, 2, 2, 2, 2] |
| LiTS | ResEnc-L | CT | [1.0, 0.77, 0.77] | 2 | [192, 192, 192] | [[1, 1, 1], [2, 2, 2], [2, 2, 2], [2, 2, 2], [2, 2, 2], [2, 2, 2]] | [1, 3, 4, 6, 6, 6] |
| SST3 | Default | CT | [5.0, 1.17, 1.17] | 2 | [40, 224, 192] | [[1, 1, 1], [1, 2, 2], [1, 2, 2], [2, 2, 2], [2, 2, 2], [2, 2, 2]] | [2, 2, 2, 2, 2, 2] |
| SST3 | ResEnc-L | CT | [5.0, 1.17, 1.17] | 2 | [56, 320, 256] | [[1, 1, 1], [1, 2, 2], [2, 2, 2], [2, 2, 2], [2, 2, 2], [2, 2, 2], [1, 2, 2]] | [1, 3, 4, 6, 6, 6, 6] |
| MAMA | Default | Z-Score | [2.0, 0.7, 0.7] | 2 | [56, 192, 192] | [[1, 1, 1], [1, 2, 2], [2, 2, 2], [2, 2, 2], [2, 2, 2], [1, 2, 2]] | [2, 2, 2, 2, 2, 2] |
| MAMA | ResEnc-L | Z-Score | [2.0, 0.7, 0.7] | 2 | [80, 256, 256] | [[1, 1, 1], [1, 2, 2], [2, 2, 2], [2, 2, 2], [2, 2, 2], [2, 2, 2], [1, 2, 2]] | [1, 3, 4, 6, 6, 6, 6] |
| SBM | Default | Z-Score | [1.0, 0.94, 0.94] | 2 | [112, 160, 128] | [[1, 1, 1], [2, 2, 2], [2, 2, 2], [2, 2, 2], [2, 2, 2], [1, 2, 2]] | [2, 2, 2, 2, 2, 2] |
| SBM | ResEnc-L | Z-Score | [1.0, 0.94, 0.94] | 3 | [160, 192, 160] | [[1, 1, 1], [2, 2, 2], [2, 2, 2], [2, 2, 2], [2, 2, 2], [2, 2, 2]] | [1, 3, 4, 6, 6, 6] |
| Atlas22 | Default | Z-Score | [1.0, 1.0, 1.0] | 2 | [128, 128, 128] | [[1, 1, 1], [2, 2, 2], [2, 2, 2], [2, 2, 2], [2, 2, 2], [2, 2, 2]] | [2, 2, 2, 2, 2, 2] |
| Atlas22 | ResEnc-L | Z-Score | [1.0, 1.0, 1.0] | 2 | [160, 224, 192] | [[1, 1, 1], [2, 2, 2], [2, 2, 2], [2, 2, 2], [2, 2, 2], [2, 2, 2]] | [1, 3, 4, 6, 6, 6] |
| Word | Default | CT | [3.0, 0.98, 0.98] | 2 | [64, 192, 160] | [[1, 1, 1], [1, 2, 2], [2, 2, 2], [2, 2, 2], [2, 2, 2], [2, 2, 2]] | [2, 2, 2, 2, 2, 2] |
| Word | ResEnc-L | CT | [3.0, 0.98, 0.98] | 2 | [96, 224, 224] | [[1, 1, 1], [1, 2, 2], [2, 2, 2], [2, 2, 2], [2, 2, 2], [2, 2, 2]] | [1, 3, 4, 6, 6, 6] |

convolution operations to extract pseudo-sequences from inputs. While there are notable efforts to catalogue the seemingly 100s of Transformer-driven techniques for medical image segmentation (Shamshad et al., 2023; Xiao et al., 2023; He et al., 2023a; Li et al., 2021), some techniques have exerted considerable influence in this domain.

### B.1 VISION TRANSFORMERS IN MEDICAL IMAGE SEGMENTATION

Vision Transformers (ViTs) with global attention were used more frequently in early work. Owing to the large sizes of medical images and the prohibitive memory-cost of Transformer layers with long sequences, initial approaches used 2D medical image slices and a 2D convolutional sequence extractor couple with Transformer layers. Among the earliest approaches, TransUNet (Chen et al., 2021) utilized a Transformer block in the bottleneck of a UNet architecture which limited memory-consumption during long-range representation learning. Another work, LeViT-UNet (Xu et al., 2023) also used a noticeably similar architectural design. This architecture design was extended to 3D by the TransBTS (Wang et al., 2021) and BiTr-Unet (Jia & Shu, 2021) which utilized Transformer layers in the bottleneck of a 3D-UNet design for brain tumor segmentation. These approaches limited the length of sequences while using convolutional blocks prior to Transformers to increase the receptive field of tokens. Slightly different from this, the UNet-Transformer (Petit et al., 2021), used a Transformer block in the bottleneck while using Cross Attention modules to integrate Encoder features into the Decoder. The UTNet (Gao et al., 2021) architecture, on the other hand, used individual Transformer blocks on each spatial resolution of a UNet, while techniques such as (Xie et al., 2021b; Huang et al., 2021) used a single Transformer to jointly learn representations from multiple spatial resolutions. Another work, TransAttUnet (Chen et al., 2023), used a self-attention block in the bottleneck as a Transformer self-attention block to learn a non-local representation for 2D medical image segmentation. One of the most influential works using a Vision Transformer, UNETR (Hatamizadeh et al., 2022), used a Transformer encoder prior to directly encode a 3D volume prior to a convolutional network. TransFuse (Zhang et al., 2021) on the other hand, was 2D network which used separate branches for Convolutional and Transformer operations while merging them in upper layers.

### B.2 SWIN TRANSFORMERS IN MEDICAL IMAGE SEGMENTATION

Shifted Window Attention (Liu et al., 2021b) networks, as opposed to ViTs, compute attention within localized non-overlapping windows. Feature mixing across windows is performed by shifting windows by an offset and attention recomputation. Swin-Transformers incorporate patch merging blocks, resembling pooling in standard convolutional networks, for hierarchical representation learning, and have also found popular use in medical image segmentation. One of the popular forms

of usage is the replacement of convolutions by Swin-blocks in a UNet-like architecture. SwinUNet (Cao et al., 2022) was one of the earliest attempts to propose such an architecture. Owing to the 2D nature of the network, it benefited from transfer learning from ImageNet trained weights for improved performance on multiple tasks. DS-TransUNet (Lin et al., 2022) used a similar Swin-Transformer-based UNet architecture with explicit low and high resolution encoder branches for 2D medical image segmentation. In due course this architecture design was adopted in 3D networks. VT-UNet (Peiris et al., 2022) proposed UNet-based architecture with 3D Swin-Tranformer blocks for the segmentation of tumors in brain MRIs. nnFormer (Zhou et al., 2023) on the other hand leveraged automated architecture design similar to nnUNet (Isensee et al., 2021) to offer performance comparable to nnUNet on a plethora of 3D medical image segmentation tasks[2]. The authors also demonstrated that ensembling nnFormer and nnUNet predictions has a complimentary effect of improving overall segmentation performance. Some architectures modify concepts within the original Swin-Transformer such as D-Former (Wu et al., 2023) which used Swin blocks with local and dilated windows for representation learning for 3D medical image segmentation. Some architectures borrowed heavily from influential ViT-based approaches published in previous years. SwinBTS (Jiang et al., 2022) built on the approach of TransBTS by using a similar architecture, but using a Swin Transformer in the bottleneck (instead of a Vision Transformer). Similar to TransFuse in ViT-based networks, CTC-Net (Yuan et al., 2023) introduced a multi-branch Swin and convolutional network for effective segmentation of organs and cardiac tissue. One of the most influential works in 3D medical image segmentation using Transformers was the SwinUNETR (Hatamizadeh et al., 2021), which improved upon the architecture of the UNETR by replacing its ViT with a 3D Swin-Transformer. They demonstrated improved performance in brain tumor segmentation. In a follow-up work (Tang et al., 2022), they also demonstrated using this architecture that self-supervised pretraining on a large medical image dataset, could benefit performance in a variety of organ and pathology segmentation tasks in 3D medical image segmentation.

Table 9: **Transformer-based networks are powered by ConvNets.** Upon closer inspection, 8 out of 9 architectures make extensive use of convolutions resulting in a **high UNet-index**, ranging between 24-352% of the total parameters of a standard UNet. We see in Fig. 1, networks with such high UNet-indices show limited performance loss on complete removal of their Transformer. In comparison, Primus is a low UNet-index network with high performance, heavily using its Transformer for learning representations.

| | TransBTS | TransFuse | TransUNet | UNETR | UTNet | CoTr | nnFormer | SwinUNet | SwinUNETR | Primus |
|---|---|---|---|---|---|---|---|---|---|---|
| Input Dims | 3D | 2D | 2D | 3D | 2D | 3D | 3D | 2D | 3D | 3D |
| **Encoder** | | | | | | | | | | |
| Convolution | + | + | + | - | + | + | Re. | - | - | - |
| Transformer | - | + | - | + | + | + | + | + | + | + |
| **Decoder** | | | | | | | | | | |
| Convolution | + | + | + | + | + | + | Re. | - | + | Re. |
| Transformer | - | - | - | - | + | - | + | + | - | - |
| **Bottleneck** | | | | | | | | | | |
| Convolution | - | - | - | + | + | + | Re. | - | + | - |
| Transformer | + | - | + | - | + | - | + | + | - | - |

## B.3 Popular Transformer-based architectures for 3D medical image segmentation

A number of massively-influential Transformer architectures for medical image segmentation, regularly used as blueprints for designing newer architectures or state-of-the-art baselines. In this work, we focus on 9 such networks, where 4 of them are 2D and the remaining 5 are 3D networks, with over 24500 citations collectively in the last 5 years (see Table 10): (i) TransFuse (Zhang et al., 2021) (ii) TransUNet (Chen et al., 2021) (iii) UTNet (Gao et al., 2021) (iv) SwinUNet (Cao et al., 2022) (v) SwinUNETR (Hatamizadeh et al., 2021) (vi) CoTr (Xie et al., 2021b) (vii) nnFormer (Zhou et al., 2023) (viii) TransBTS (Wang et al., 2021) (ix) UNETR (Hatamizadeh et al., 2022) . These architectures are described in the following sections.

---

[2]While the authors claimed automated architecture design to the best of our knowledge, their repository does not feature automatic planning and creation of nnFormer architectures but just three static architecture designs.

Table 10: **Transformers are influential for medical image segmentation.** Citations over the last four years (2021-2025) as of 23.09.2025 show that Transformer-based networks are extremely popular for tasks in medical image segmentation. *∗ - summation from multiple sources of paper from main authors, (W) - Workshop paper*

| Network | n-D | Year | Venue | Citations |
|---|---|---|---|---|
| **TransFuse** | 2D | 2021 | MICCAI | 1574 |
| **TransUNet** | 2D | 2021 | arXiv | 7385 |
| **UTNet** | 2D | 2021 | MICCAI | 707 |
| **SwinUNet** | 2D | 2021 | ECCV (W) | 5594 |
| **SwinUNETR**∗ | 3D | 2022 | MICCAI (W) | 3000 |
| **CoTr** | 3D | 2021 | MICCAI | 826 |
| **nnFormer**∗ | 3D | 2023 | IEEE TIP | 1117 |
| **TransBTS** | 3D | 2021 | MICCAI | 1263 |
| **UNETR** | 3D | 2022 | WACV | 3225 |
| **Total Citations** | | | | **24691** |

### B.3.1 TRANSFUSE

TransFuse is a 2D architecture which has 2 branches which both receive the same input volume - a Transformer branch and a CNN branch. The Transformer branch uses a ViT for attention based global representation learning. The CNN branch uses convolution blocks for learning local representations. A novel BiFusion block is used to merge features from both branches at multiple equivalent spatial hierarchies and transform them into the output segmentation.

### B.3.2 TRANSUNET

TransUNet is a 2D architecture which was designed to merge the strengths of ViTs and CNNs. The architecture follows a UNet structure where the ViT is embedded in the bottleneck of the architecture, with a convolutional encoder and decoder. The convolutional encoder extracts deep features for the Transformer to learn global dependencies, which the decoder reincorporates into its convolutional blocks. The positioning of the ViT after multiple downsamplings allows it to learn features while limiting sequence length and consequently minimizes memory consumption of the Transformer.

### B.3.3 UTNET

The UTNet is a 2D architecture that incorporates customized attention layers alongside standard residual convolutional blocks. The architecture proposes a custom attention layer that uses downsampled keys and values (while Queries stay in high resolution) to efficiently compute attention in encoder blocks. It performs similarly in the decoder block in cross-attention settings with the high resolution skip feature being treated as the query and the low-resolution features from lower spatial hierarchies being treated as the key and value. This architecture allows their Transformer block to be interleaved with convolution blocks.

### B.3.4 SWINUNET

SwinUNet was proposed as a 2D architecture which uses a sequence of Swin-Transformer blocks instead of standard Convolutional blocks in a UNet architecture. The ability for Swin-Transformers to maintain spatial structure post-tokenization allows them to be seamlessly treated like convolutional blocks, in a UNet backbone architecture. SwinUNet uses such an architecture alongside patch merging layers for downsampling and patch expansion layers for upsampling.

### B.3.5 SWINUNETR

SwinUNETR is a 3D architecture that leverages 3D Swin-Transformer blocks to efficiently learn scalable features. The Swin Transformer enables this architecture to benefit from attention while

localizing it to windows. The features from a succession of Swin Transformer blocks are hierarchically integrated into a convolutional encoder of a UNet-styled architecture via skip connections. The convolutional decoder subsequently transforms these features into the segmentation output.

### B.3.6 CoTr

CoTr or Co-Transformer is a 3D architecture which uses a Deformable Transformer in between a convolutional encoder and decoder. The Transformer incorporates features from multiple spatial hierarchies of the encoder for representation learning. The deformable attention mechanism allows for the learning of these representations at lower computational overheads by focusing on a limited number of key points. These features are subsequently passed to the decoder which transforms them into the segmentation mask.

### B.3.7 nnFormer

nnFormer was proposed as a family of 3D segmentation models built on top of the nnU-Net Isensee et al. (2021) framework. The architecture defined 3 regions of a UNet-like architecture - encoder, bottleneck and decoder. The encoder and the decoder used Swin Transformers to efficiently learn features at high spatial resolutions. The bottleneck region has global attention layers which are enabled by the small feature resolution deeper in the network. Downsampling and upsampling layers are implemented via strided convolutions and strided transposed convolutions respectively.

### B.3.8 TransBTS

The TransBTS is a 3D architecture which can be seen as an analog to the 2D architecture TransUNet. The network leverages a 3D ViT in the bottleneck of a 3D UNet architecture. The encoder extracts 3D representations while downsampling the input volume for the Transformer to extract global features without unreasonably increasing sequence length. These features are merged back into the network via the convolutional decoder.

### B.3.9 UNETR

The UNETR is a 3D architecture which incorporates a 3D Vision Transformer (enabled by 3D tokenization of input volumes) into a UNet architecture, thereby enabling the learning of both global and local features for the volumetric segmentation of medical images. The representations learnt by the Transformer are hierarchically incorporated via skip connections into corresponding levels of the UNet encoder, thereby enabling learning of representations at multiple scales. The Transformer models global context while the UNet provides a backbone for standard local feature based representation learning.

## C Extended Primus Results

Due to limited space in the main manuscript, we provide additional results on Primus development in Appendix C.1, provide the full results of the tokenization and input patch size ablation in Appendix C.3 and provide additional experiments trying to maintain high resolution tokens while keeping the input patch size high in Appendix C.4.

### C.1 Additional Primus Development

To refine our Primus architecture, we systematically evaluated a series of modifications, to understand the impact of various hyperparameters and design choices. The table presented in Table 11 details the results of these ablation studies, where a reference configuration with its performance is listed, followed by alternative configurations that ablate changes of this value.

We denote that in these experiments certain configurations yielded marginal improvements, yet they were ultimately not included into Primus configuration due to their limited impact. By excluding these changes we kept the configuration of Primus minimal, which reduced potential points of failure that may lead to e.g. instability on other datasets. An example of this can be seen in the case

Table 11: **Extended list of changes evaluated during development.** Ablations highlight a reference configuration and its reference value together with their performance*, with ablation experiments following that ablate changes to this value in the following rows. Moreover, despite some configurations being slightly better, we denote that these changes could be rejected due to their potentially minor influence. Hence, it was opted to exclude these changes, instead of including them to simplify the architecture and training configuration, leading to fewer potential failure points. An example of this change was the exclusion of *'Drop Attention'* which showed slight improvements but was rejected. *Embed. Dim.: Embedding Dimensions; LS: Layer Scale; PAN: Post Attention Normalization; LPe: Learnable Positional Embedding; w/o: without; w/: with; FOV: Field-of-View; *: Asterix denotes a prior version of Primus-M which suffered from a permutation error in its 3D RoPE embeddings affecting all datasets but LiTS in this table. However, all suffered equally, so findings should be reliable.*

| Configurations | | Dice Similarity Coefficient (in %) | | | | |
|---|---|---|---|---|---|---|
| Reference Configuration | Reference Value/Changed Value | ACDC | AMOS22 | KiTS23 | LiTS | Avg. |
| Eva02-MLP*; w/o DropPath; w LPe; w 3D RoPe | 0 Register Tokens | 92.28 | 87.45 | 88.16 | 82.36 | 87.56 |
| | + 1 Register Token | 91.91 | 87.55 | 87.85 | 82.60 | 87.48 |
| | + 2 Register Tokens | 92.37 | 87.39 | 87.90 | 82.11 | 87.44 |
| | + 4 Register Tokens | 92.51 | 87.69 | 88.20 | 81.11 | 87.38 |
| | + 8 Register Tokens | 92.32 | 87.58 | 87.60 | 80.86 | 87.09 |
| Eva02-MLP*; w/o DropPath; w LPe; w 3D RoPe | Embed. Dim. 864 | 92.28 | 87.45 | 88.16 | 82.36 | 87.56 |
| | Embed. Dim. 432 | 92.10 | 87.61 | 88.74 | 81.02 | 87.36 |
| | Embed. Dim. 1296 | 92.29 | 87.18 | 81.88 | 00.00 | 65.33 |
| Primus-M*; w/o LS; w/o PAN; (red in Table 3) | Drop Path 0.2 | 92.68 | 87.98 | 88.87 | 82.89 | 88.11 |
| | Drop Path 0.1 | 92.62 | 87.71 | 87.25 | 82.70 | 87.57 |
| | Drop Path 0.3 | 92.68 | 88.04 | 88.41 | 81.52 | 87.66 |
| | Drop Path 0.4 | 92.60 | 87.70 | 88.62 | 81.04 | 87.49 |
| | Drop Path 0.5 | 92.60 | 87.67 | 88.55 | 82.06 | 87.72 |
| | Drop Path 0.6 | 92.79 | 87.82 | 88.44 | 80.24 | 87.32 |
| Primus-M*; w/o LS; w/o PAN; w/Drop Path 0.2 (red in Table 3) | 0 Register Tokens | 92.68 | 87.98 | 88.87 | 82.89 | 88.11 |
| | + 4 Register Tokens (again) | 92.45 | 87.97 | 88.65 | 82.78 | 87.96 |
| Primus-M*; w/o LS; w/o PAN; w/Drop Path 0.2 (red in Table 3) | - | 92.68 | 87.98 | 88.87 | 82.89 | 88.11 |
| | + Drop Projection 0.2 | 3.31 | 0.07 | 0.00 | 2.79 | 1.54 |
| | + Drop Attention 0.2 | 92.49 | 88.09 | 88.60 | 84.30 | 88.37 |
| | + Drop Proj. 0.2 & Drop Att. 0.2 | 2.32 | 0.06 | 2.60 | 0.71 | 1.42 |
| Primus-M*; w/o LS; w/o PAN; w/Drop Path 0.2 (red in Table 3) | 3D-RoPE FOV 100 | 92.68 | 87.98 | 88.87 | 82.89 | 88.11 |
| | 3D-RoPE FOV 75 | 92.61 | 88.00 | 86.75 | 82.40 | 87.44 |
| | 3D-RoPE FOV 50 | 92.64 | 88.26 | 88.37 | 83.27 | 88.14 |
| | 3D-RoPE FOV 150 | 92.64 | 87.60 | 87.92 | 81.14 | 87.33 |
| | 3D-RoPE FOV 200 | 92.29 | 87.29 | 87.97 | 82.66 | 87.55 |
| Primus-M* (green in Table 3) | Light-weight Decoder | 92.86 | 88.12 | 88.28 | 82.42 | 87.92 |
| | + 3x3x3 Conv. per Transposed Conv. | 92.07 | 87.45 | 85.49 | 79.30 | 86.08 |

of *'Drop Attention'* and the 3D-RoPE *'Field-of-View'* changes, which showed minor performance improvements but which were ultimately rejected.

**Register tokens** It can be seen that including minor numbers of register tokens had negligible effects, while increasing their count to eight decreased performance slightly. This suggests that excessive register tokens may introduce redundant representations that do not contribute meaningfully to segmentation quality. This may originate from the large amount of overall tokens. Our sequence length, depending on the dataset, was about 13k tokens, hence the likelihood of some being uninformative and could serve as registers is very high. Given the large amount of tokens, it is rather interesting to find that the inclusion of a minor amount of registers had such a large impact on training behavior at all.

**Drop Path** A critical parameter examined was Drop Path, where we varied the drop rate from 0.1 to 0.6. While moderate Drop Path rates (e.g., 0.2–0.3) appeared to improve generalization, values beyond 0.5 had diminishing benefits, suggesting that excessive stochastic regularization may disrupt the overall learning process. Hence, a final value of 0.2 was used in Primus.

**3D-RoPE Field-of-View** When exploring the impact of modifications to the 3D Rotary Position Embedding (3D-RoPE) we explored modifications to the Field-of-View (FOV) choice. This parameter steers the frequency of rotation, with lower FOV values leading to a faster decay and higher FOV values leading to a slower decay. With lower FOV values it is more difficult for the model to learn long-range dependencies, while lower values enable easier learning of long-range dependencies. In the experiments conducted, it can be observed that reducing the FOV to 50% retained compet-

itive performance, while increasing it beyond 150% resulted in slight degradation across datasets. However, despite the minor improvements, the increase is not consistent for e.g. the 75% FOV not indicating performance benefits. Moreover, performance on LiTS and AMOS22 increased slightly, while performance on KiTS and ACDC decreased, indicating that this change may not generalize well across datasets. Hence adaptations of the RoPE FOV were not included.

**Light-weight vs. Larger Convolutional Decoder**    Finally, we assessed the impact of our initial design choice—a light-weight decoder—by exploring the effects of increasing its size. Specifically, we introduced additional convolutional layers between the transposed convolutions during the up-sampling process. Our results indicate that the light-weight decoder maintains strong performance, whereas incorporating 3×3×3 convolutions into the transposed convolutions degrades results on all datasets consistently. This suggests that the inclusion of excessive convolutional operations in the decoder stage introduces unnecessary computational overhead and potentially inhibits the Transformer from learning good representations.

## C.2    Tokenization and Iterative Patch Embedding

The Primus architecture uses a single, default projection (DP) of $8 \times 8 \times 8$ voxels to a token. This projection is required to capture all relevant signals and represents a potential bottleneck in case of failure. Consequently, we evaluate whether this projection style is sufficient to capture all the required signals of the medical image by introducing different tokenization schemes. Specifically, we choose to compare four patch-embedding schemes, with increasing convolutional footprint.

1. **Iterative Projection (IP):** Instead of directly projecting from 8x8x8 pixels to our embedding dimension, we introduce three iterative stages of projecting down. We achieve this by using Convolutions with kernel size $2 \times 2 \times 2$ and stride $2 \times 2 \times 2$, using a `Conv-Norm-Act` structure. Lastly, we use a single Convolution with $1 \times 1 \times 1$ kernel to project to the desired embedding dimension. By chaining three of the `Conv-Norm-Act` blocks together (channel depths: [32, 64, 128] respectively) followed by the projection, this patch embedding yields the same amount of tokens as previously, with tokens still having no overlap between each other.

2. **Convolutional Downsampling (CD):** We introduce an initial `Conv-Norm-Act` Stem with kernel size $3 \times 3 \times 3$ projecting the input to channel depth 32. This is followed by three consecutive blocks as in the 'Iterative Projection', but instead of $2 \times 2 \times 2$ kernel size we increase the kernel size to $3 \times 3 \times 3$. This increases expressivity of the kernels but at the same time, introduces a slight overlap of the field-of-view between each token, due to the stride being 2. These blocks are identical to the ones used in the default nnU-Net encoder.

3. **Minimal Residual Downsampling (MRD):** We follow the same block structure as in the Convolutional Downsampling, but instead of using plain `Conv-Norm-Act` block, we use a residual basic block, like in the ResEnc-L architecture and as visualized on the right in Fig. 3. This increases token overlap and UNet Index further, but allows creating more semantic tokens. This is the *Iterative Tokenizer* used in PrimusV2 (Section 3.2)

4. **Large Residual Downsampling (LRD):** We expand the amount of residual blocks from before from [1-1-1] per resolution to [1-2-3] – the stem remains the same. This increases the amount of blocks per stage substantially and leads to large field-of-view overlaps between tokens, increasingly becoming like the encoder of the Residual Encoder UNet, which starts with [1-3-4] blocks (The full ResEnc-L Blocks per stage are visualized in Table 8, commonly being [1-3-4-6-6-6]).

We report the DSC values on fold 0 of our development datasets in Table 12. It can be observed that increasing the patch-embeddings from the Default Projection, as used in Primus , to Iterative Projections, to Convolutional Downsampling and Residual Downsampling, increases the overall performance of the Transformer. However, at the same time, this increase translates to the architecture increasing in overall performance without the Transformer present – similar to our findings of existing architectures in (Section 2). In particular, when using the LRD patch embedding, performance differences reduces to less than three DSC points, and on ACDC even exceeds the full architecture and ResEnc-L itself. As we want a transformer architecture, where the Transformer measurably contributes to overall performance, we choose to use the Minimal Residual Downsampling as the Patch Embedding for PrimusV2, as it yields the highest performance, while still exhibiting a considerable performance difference of five DSC points on Average with the transformer removed.

Table 12: **Iterative tokenizers improve performance.** We report DSC values of fold 0 of the development datasets for various versions of Patch Embeddings, with and without replacing the transformer with an identity. *PE: Patch Embedding; w/TR: with Transformer; DP: Default Projection; IP: Iterative Projection; CD: Convolutional Downsampling; MRD: Minimal Residual Downsampling; LRD: Large Residual Downsampling*

| PE/Model | w/TR | ACDC | AMOS22 | KiTS23 | LiTS | Avg. |
|---|---|---|---|---|---|---|
| nnUNet def. | - | 92.43 | 88.75 | 86.22 | 82.48 | 87.47 |
| nnUNet ResEnc-L | - | 93.05 | 89.65 | 89.16 | 82.89 | 88.69 |
| DP | ✗ | 19.45 | 03.02 | 15.88 | 40.76 | 19.78 |
| DP (Primus-M) | ✓ | 92.73 | 87.89 | 88.11 | 82.89 | 87.91 |
| IP | ✗ | 38.51 | 25.05 | 27.61 | 54.05 | 36.30 |
| IP | ✓ | 92.62 | 89.19 | 88.68 | 84.01 | 88.63 |
| CD | ✗ | 76.79 | 54.61 | 55.79 | 69.12 | 64.08 |
| CD | ✓ | 92.68 | 89.24 | 89.30 | 83.28 | 88.62 |
| MRD | ✗ | 91.89 | 81.56 | 82.69 | 79.77 | 83.98 |
| MRD (PrimusV2-M) | ✓ | 92.81 | 89.34 | 88.57 | 84.91 | 88.91 |
| LRD | ✗ | 93.17 | 87.00 | 86.61 | 83.14 | 87.48 |
| LRD | ✓ | 93.10 | 89.34 | 89.26 | 84.72 | 89.11 |

## C.3 FULL SMALLER TOKENIZATION AND INPUT PATCH SIZE RESULTS

In this section, we analyze the impact of reducing the token size from $[8 \times 8 \times 8]$ to $[4 \times 4 \times 4]$ while simultaneously halving the input patch size, ensuring that the overall sequence length remains constant. To disentangle the effects of the input patch size and the token size, we train an additional baseline with the same original token size of $[8 \times 8 \times 8]$ but halved input patch size. We provide results of fold 0 only, as re-training all 5 folds of the cross-validation would induce a significant computational overhead, hence all reference values of e.g. nnU-Net or CoTr just feature fold 0. The results are presented in Table 13.

It can be seen that decreasing input patch size and token size has vastly different effects, depending on the dataset they are trained on. For datasets where reducing the input patch size significantly affects segmentation performance—such as AMOS22, KiTS, and LiTS—increasing the token size to $[4 \times 4 \times 4]$ helps recover some of the lost performance. However, this recovery is only partial and does not fully compensate for the degradation caused by the smaller input patch size. This suggests that for these datasets, input patch size, and simultaneously the availability of a more global context, plays a critical role to reach high performance. Subsequently, reducing it without any compensatory adjustments can be detrimental to overall performance and is likely not recommendable. When introspecting predictive behavior on KiTS23, we observe that Primus-M/4 with halve input patch size shows approximately the same amount of mean False Negative Voxels per case w.r.t Primus-M/8 with full input patch size, 8466 vs 8360, while making more than twice the amount of mean false positive errors per case, 29016 vs 13645. This indicates, that the lack of context leads to confusing areas not part of the kidney as kidney. This lack of orientation likely extends to other tasks, e.g. the abdominal segmentation tasks, where global understanding of locality is crucial.

On the other hand, there are datasets where the reduction in input patch size has minimal or even positive effects on segmentation performance, such as ACDC, SBM, and Atlas22. For these datasets a decrease in token size leads to an overall improvement in performance. This indicates that in certain cases where understanding of global position is not as crucial, smaller token sizes improve local positioning and hence improve overall segmentation accuracy. Particularly, the Stanford Brain Metastases (SBM) dataset features small brain metastases. The main difficulty of this task is to identify hyperintense brain metastases lesions and disambiguate them from vessels, which similarly appear hyper-intense due to the contrast agent in the bloodstream. Subsequently, the difficulty lies in local identification. If the hyperintensity has a clear beginning and ending it is likely a lesion, but if it has a long winding structure exiting the field of view it likely is a vessel. Hence, the overall reduction in field-of-view induces a positive influencing locality bias, while the further reduction in token

Table 13: **Influence of decreasing token size and input patch size.** Reducing the token size from 8 to 4 and simultaneously reducing input patch size by half to maintain overall sequence length shows different effects on different datasets. On datasets where decreasing the input patch size has a strong effect on segmentation performance* (e.g. AMOS22, KiTS and LiTS) it can be observed that increasing the token size to $[4 \times 4 \times 4]$ recovers some performance but cannot offset the prior loss. On datasets where the effect of reducing input patch size is minimal or even positive for performance (e.g. ACDC, SBM or Atlas22) the decrease in token size increases absolute performance. Hence, we find the choice of input patch size and token size to be crucial factors, that may need manual adjustments depending on the dataset Primus is applied to. Moreover, we want to highlight that while reducing token size may positively influence overall segmentation performance, it simultaneously leads to a large sequence length which can reduce its applicability for multi-modal applications. The Primus-X/8 configurations represent the default Primus configuration. The /8 and /4 is added to indicate the token patch size used. *IPS: Input Patch Size, *: Asterix denotes a prior version of Primus-M which suffered from a permutation error in its 3D RoPe embeddings affecting all datasets but LiTS in this table. However, all suffered equally, so findings should be reliable.*

| Trainer | IPS | ACDC | AMOS22 | KiTS23 | LiTS | SST3 | MAMA | SBM | Atlas22 | Word |
|---|---|---|---|---|---|---|---|---|---|---|
| | | | | | | Dice Similarity Coefficient (DSC) on Datasets | | | | |
| nnUNet def. | Full | 92.43 | 88.75 | 86.22 | 82.48 | 90.25 | 78.41 | 68.52 | 62.31 | 82.75 |
| nnUNet ResEnc-L | Full | 93.05 | 89.65 | 89.16 | 82.89 | 90.32 | 79.68 | 70.97 | 62.46 | 85.73 |
| nnUNet def. | Half | 92.65 | 82.99 | 51.33 | 68.32 | 90.50 | 69.56 | 71.63 | 54.27 | 80.24 |
| nnUNet ResEnc-L | Half | 93.55 | 88.62 | 87.01 | 79.72 | 90.31 | 77.36 | 72.46 | 63.56 | 83.79 |
| CoTR | Full | 90.81 | 88.36 | 84.42 | 81.71 | 89.22 | 77.47 | 66.47 | 61.52 | 83.23 |
| nnFormer | Full | 92.61 | 82.33 | 76.35 | 78.88 | 88.42 | 68.31 | 71.45 | 60.90 | 83.70 |
| SwinUNETR | Full | 91.36 | 81.75 | 77.22 | 77.13 | 87.54 | 76.23 | 66.99 | 61.09 | 80.42 |
| UNETR | Full | 89.80 | 64.67 | 78.08 | 75.56 | 84.55 | 73.39 | 53.67 | 53.39 | 73.01 |
| Primus-S/8* | Full | 92.46 | 87.47 | 86.76 | 82.89 | 88.25 | 76.57 | 58.98 | 61.99 | 84.06 |
| Primus-B/8* | Full | 92.70 | 87.87 | 86.83 | 83.16 | 88.50 | 75.47 | 58.16 | 61.73 | 84.16 |
| Primus-M/8* | Full | 92.86 | 88.12 | 88.28 | 82.42 | 88.64 | 75.67 | 57.56 | 61.44 | 84.31 |
| Primus-L/8* | Full | 92.71 | 88.60 | 88.64 | 83.00 | 88.46 | 76.12 | 55.49 | 59.60 | 84.01 |
| Primus-S/8* | Half | 92.82 | 81.07 | 80.63 | 76.00 | 87.07 | 71.53 | 60.92 | 57.64 | 80.94 |
| Primus-B/8* | Half | 92.69 | 83.59 | 82.98 | 78.78 | 87.42 | 71.51 | 62.33 | 56.01 | 82.56 |
| Primus-M/8* | Half | 92.80 | 83.82 | 81.19 | 78.60 | 87.73 | 71.24 | 64.02 | 54.30 | 82.07 |
| Primus-L/8* | Half | 92.98 | 81.65 | 79.58 | 76.56 | 87.41 | 71.99 | 62.55 | 50.83 | 81.68 |
| Primus-S/4* | Half | 93.25 | 85.61 | 83.45 | 79.98 | 88.69 | 76.85 | 68.31 | 61.27 | 83.31 |
| Primus-B/4* | Half | 93.17 | 86.96 | 81.89 | 80.19 | 88.76 | 77.81 | 69.82 | 62.00 | 84.16 |
| Primus-M/4* | Half | 93.17 | 87.26 | 83.70 | 80.84 | 88.99 | 76.72 | 69.72 | 62.79 | 83.64 |
| Primus-L/4* | Half | 93.04 | 87.74 | 84.79 | 79.97 | 88.75 | 76.36 | 67.05 | 61.54 | 83.68 |

size, improves the fine-grained localization of the small lesion, boosting segmentation performance further.

These findings highlight that input patch size and token size are critical hyperparameters that must be carefully selected based on a dataset's characteristics. At present, there is no universally optimal configuration that ensures out-of-the-box generalization, necessitating manual adjustments when applying Primus to different datasets. Furthermore, while naively reducing token size can enhance segmentation performance in some cases, it comes at a significant cost. The resulting increase in the total number of tokens needed to create embeddings for an entire case leads to a longer overall sequence length. This, in turn, reduces the feasibility of using such models for multi-modal applications, as longer sequences impose substantial memory and computational constraints. In contrast, our usage of the iterative tokenizer in PrimusV2 (Section 3.2) does significantly improve performance on a number of datasets, including SBM with small lesions as mentioned earlier.

In summary, our analysis underscores the importance of balancing input patch size, token size, sequence length, as well as the overall design of the tokenizer to optimize segmentation performance across diverse datasets. Future work could focus on developing strategies to dynamically adjust these parameters based on dataset properties, improving both generalization and computational efficiency.

Due to the findings of the efficacy of smaller tokens and the need for larger context we also conducted some initial, naive experiments that tried to merge these two worlds, by introducing token sparsity during training time, which we detail in Appendix C.4.

## C.4 LARGE CONTEXTS AND SHORTER SEQUENCES

The results of the ablation of reduced token size $[8 \times 8 \times 8] \rightarrow [4 \times 4 \times 4]$ and full input patch and halved input patch size indicate that having context can be helpful and that smaller tokens improve performance similarly. Hence, we wonder if we can achieve the best of both worlds by decreasing the token size, while maintaining the sequence length *at training time*, and providing full context at inference. Due to Primus working in token space this can be achieved through various token masking strategies where we, instead of applying minor amounts of token drop-out, remove the majority (87.5%) of all tokens. As the structure and amount of sparsity may be the determining factor we evaluate the following schemes:

1. **Structured masking:** To maintain spatial structure and long context ranges, we pick a random axis and keep as many contiguous slices along that axis to reach the desired sparsity level. To reach a consistent amount of tokens, additional tokens are chosen from the neighboring slices including them partially. This type of masking allows the model to learn long context range cues across each direction, however, it doesn't see long context range cues from multiple directions which may limit it's applicability

2. **Random masking:** Random masking samples a fixed amount of tokens from the entire volume randomly. Due to the high amount of sparsity, this will lead to many hardly connected tokens, and possibly a very difficult task to solve, however it will allow learning global contexts.

As the masking ratio of 87.5% is rather excessive in the random masking setting, we decided to first evaluate the performance degradation on $[8 \times 8 \times 8]$ token size as this allows us to explore lower levels of sparsity, which would otherwise lead to too long sequences and too much VRAM consumption to evaluate on the smaller token sizes. Results are presented in Table 14.

Introspecting the results, it can be observed that the random masking strategy drops in performance quickly and consistently. Particularly for the sparsity level required to maintain sequence length (87.5%) average performance on AMOS22 and ACDC dropped to 87.44 from 90.32, which we deemed unrecoverable, hence this approach was discarded.

Further, the structured masking approach lost an absolute of about 1.3 DSC points for 85% sparsity, which would result in a slightly longer sequence. Particularly for AMOS22, a dataset where we previously showed that global context is important (Appendix C.3), one still observes a substantial decrease in performance, putting to question the efficacy of the masking approach. We hypothesize that this may be to two reasons: i) When masking out a large portion of the input, the number of visible classes in each batch will decrease substantially. Depending on the size of the target structure, one may even end up with patches without any visible foreground classes, as our structured masking is mask agnostic. This will negatively impact sampling efficiency and training convergence and would necessitate improvements. ii) The masking strategy employed limits visibility to contiguous slices, be they axial, sagittal or coronal. Subsequently, the Transformer experiences a shift during inference when all contexts are visible at the same time, which may degrade performance. Further, the lack of permanent availability of long context cues may lead to decreased emphasis on learning these relations and subsequently less global reasoning in the model.

Even if we would be able to achieve parity with smaller tokens one would still run into the undesirable effect that conducting using the full input patch size with smaller token size leads to excessive inference times. While we were able to fit this into the VRAM of an A100 40GB GPU, the throughput decreased drastically due to the 8x longer sequence and the 64x times more costly self-attention. Hence, we decided not to pursue this direction further and leave this problem open. While our current experiments yielded negative results, we believe unlocking smaller token sizes with larger input patch sizes is a research direction that can allow Transformers to exceed convolutional neural networks, e.g. through linear attention paradigms or through token aggregation approaches which can reduce the large amount of redundant tokens in the sequence, effectively shortening the sequence length.

Table 14: **Effects of random and structured masking.** To evaluate the feasibility of reducing token size to $[4 \times 4 \times 4]$ we measure the effect of random various random masking experiments as well as the effect of structured masking on $[4 \times 4 \times 4]$ token size. It can be observed that random masking performance* degrades rapidly for larger sparsity levels. Due to requiring about 87.5% sparsity to maintain input patch size (and simultaneously sequence length) when reducing token size from 8 to 4, this approach is deemed infeasible. Structured masking, tested on token size 4 directly, shows improvements on ACDC, but substantial decreases on AMOS22, which is a dataset for which the global context was previously shown to be important. Hence, the single axis structured masking seems to be infeasible as well, hence was dropped in development for Primus. *: *Asterix denotes a prior version of Primus-M which suffered from a permutation error in its 3D RoPe embeddings affecting all datasets but LiTS in this table. However, all suffered equally, so findings should be reliable.*

| Token Size | Masking style | Sparsity [%] | ACDC* | AMOS22* | Average* |
|---|---|---|---|---|---|
| $[8 \times 8 \times 8]$ | Baseline configuration | | 92.51 | 88.13 | 90.32 |
| $[8 \times 8 \times 8]$ | Random | 13 | 92.28 | 88.29 | 90.29 |
| $[8 \times 8 \times 8]$ | Random | 25 | 92.10 | 88.04 | 90.07 |
| $[8 \times 8 \times 8]$ | Random | 37 | 91.70 | 87.69 | 89.69 |
| $[8 \times 8 \times 8]$ | Random | 50 | 91.46 | 87.32 | 89.39 |
| $[8 \times 8 \times 8]$ | Random | 63 | 91.12 | 87.03 | 89.08 |
| $[8 \times 8 \times 8]$ | Random | 75 | 91.02 | 86.20 | 88.61 |
| $[8 \times 8 \times 8]$ | Random | 87 | 90.71 | 84.17 | 87.44 |
| $[4 \times 4 \times 4]$ | Structured | 85 | 92.65 | 85.42 | 89.04 |
| $[4 \times 4 \times 4]$ | Structured | 87.5 | 92.48 | 85.14 | 88.81 |
| $[4 \times 4 \times 4]$ | Structured | 92.5 | 92.18 | 82.84 | 87.51 |

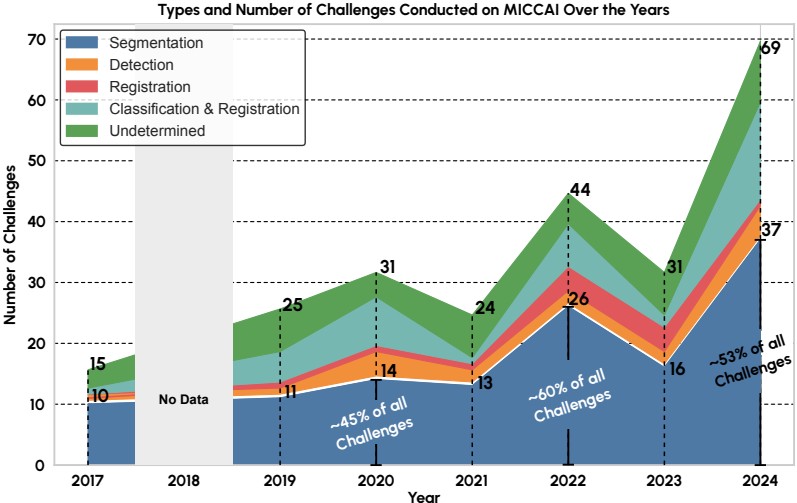

Figure 6: MICCAI challenges categorized by their task. Since a long time at least 50% of challenges only focus on semantic segmentation with other tasks being significantly less represented.

# D DATA IN MEDICAL IMAGE ANALYSIS

## D.1 SEMANTIC SEGMENTATION CHALLENGES AT MICCAI

A significant testament to the importance of semantic segmentation in the medical imaging community is reflected in the annual MICCAI (Medical Image Computing and Computer Assisted Intervention) conference. A vast majority of challenges and competitions at MICCAI revolve around semantic segmentation. Fig. 6 illustrates the dominance of semantic segmentation challenges at the MICCAI conference, highlighting the central role it occupies in advancing the field of medical image analysis. In summary, semantic segmentation serves as a cornerstone in 3D medical image analysis, particularly in the context of MRI and CT data. Its native representation, support in diagnosis and treatment planning, and contributions to personalized medicine are instrumental in reshaping healthcare. The synergy between computer vision and medical imaging, driven by semantic segmentation, holds promise for improving patient care and catalyzing transformative advancements in 3D medical image segmentation.

## D.2 THE DATA 'CHASM' BETWEEN NATURAL AND MEDICAL IMAGES

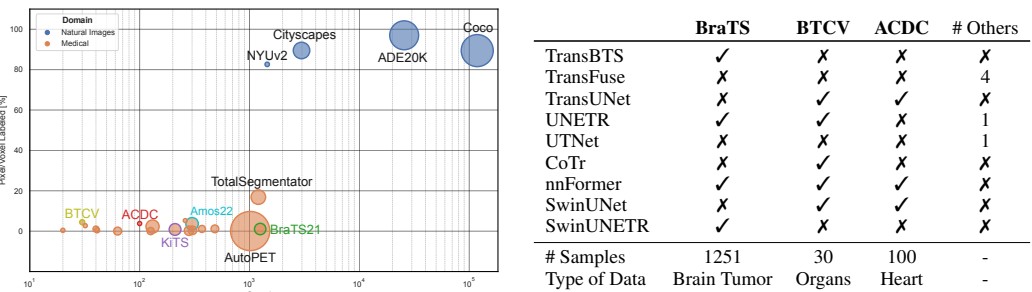

Figure 7: **Medical image segmentation datasets are significantly smaller and sparsely-labeled compared to their natural image counterparts.** Our dataset visualization (Left) illustrates this chasm by the *Average Percentage of Image/Volume Labeled* vs. *Number of Samples* of datasets from both domains. Radii visualizes pixel/voxels over the whole dataset. However, the original evaluation of our 9 Transformer-based models (Right) shows repeated usage of these same small datasets.

Transformer architectures are difficult to train from scratch on small scale datasets, regardless of the domain (Liu et al., 2021a). Therefore pre-training on large datasets is preferred for large Transformer networks even in the natural image domain (Dosovitskiy et al., 2021). The datasets commonly used for this are ImageNet1k with 1.3M images (Russakovsky et al., 2015), ImageNet21k with 14M images (Sun et al., 2017) or even larger proprietary datasets like JFT-300M with 303M images. The realm of medical image segmentation stands in stark contrast to this. Due to the lack of prominent, monolithic architectures and huge datasets that work well for the heterogeneous downstream tasks, almost all models are trained from scratch. The datasets are commonly of small scale, featuring only 10s or 100s of samples (Litjens et al. (2017), Li et al. (2021)). Complicating it further, the samples tend to be sparsely annotated, containing only annotations for a few classes of interest – while natural imaging segmentation datasets tend to be largely fully-labeled.

More recently the TotalSegmentator dataset (Wasserthal et al., 2022), AbdomenAtlas (Li et al., 2024; Qu et al., 2024), FLARE 2023 (Ma et al., 2024) and multi-dataset training (Ulrich et al., 2023) have taken a step in the right direction, tackling the data-sparsity that plagues the medical image domain. We demonstrate this severe chasm between datasets of the medical and natural image segmentation domain in Fig. 7 (Left) by contrasting them by their *number of samples* and their *average fraction of annotated foreground* in each sample. The low dataset size and annotation-sparsity pose substantial difficulties when training architectures in the medical domain. While some Transformer backbones of TransFuse, SwinUNet and TransUNet are pre-trained on ImageNet, the majority of performant architectures – UNETR, CoTr, SwinUNETR, nnFormer, UTNet and TransBTS – train from scratch, with some being trained on BTCV, a dataset comprised of 30 samples. This highlights that data

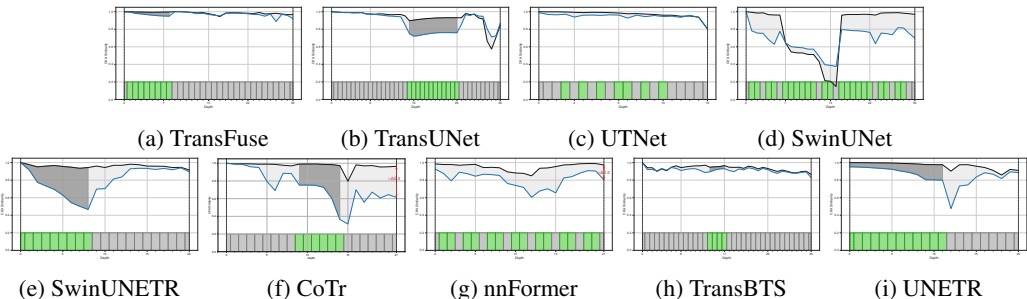

(a) TransFuse      (b) TransUNet      (c) UTNet      (d) SwinUNet

(e) SwinUNETR     (f) CoTr     (g) nnFormer     (h) TransBTS     (i) UNETR

Figure 8: **Impact of Transformer blocks on learned representations across different architectures.** We measure the representational similarity using centered kernel alignment (CKA) between multiple training runs of the same Transformer architecture (black) and between a Transformer architecture and its variant where Transformer blocks are replaced with identity mappings (blue). The gray-shaded region highlights the gap between these two similarity measures, indicating the extent to which Transformer blocks alter learned representations. For six out of nine architectures, the final output representations remain nearly identical, suggesting minimal impact from the presence of Transformer blocks. Green-highlighted layers at the bottom denote Transformer blocks within each architecture.

size restrictions native to the medical image segmentation domain are a roadblock to outperforming CNNs with Transformer-based architectures.

# E   EFFECT ON LEARNED REPRESENTATIONS

While absolute performance changes may be the first indicator of a lack of influence of the Transformer blocks, representation learning may still be influenced by the presence of the Transformer blocks. To this end, the representational similarity of a trained model with the identity-replacement of the Transformer blocks and one trained without this identity replacement (blue) is measured. This allows us to quantify to what degree the learned representations are influenced by the Transformer. To get a reference of normal variations of representational similarity, we compare it against different random seed training runs of the Transformer architectures without any changes (black), illustrated in Fig. 8.

As similarity measure, we use centered kernel alignment (CKA) (Kornblith et al., 2019). More precisely, we use minibatch CKA (Nguyen et al., 2020) which utilizes unbiased HSIC of Song et al. (2012). We re-use three seeds of the Transformer and replaced-Transformer architectures of Section 2.1 to extract representations. For explicit experiment details on where representations are extracted and which data was used, we refer to Appendix E.1. The following can be observed:

1. **No change during Transformer blocks:** 3 out of 9 networks (TransBTS, UTNet and TransFuse) show little to no change in learned representations at the Transformer blocks when they are removed. This highlights a severe architectural issue where Transformer blocks are completely ineffective at learning useful representations.

2. **No change at output:** 6 out of 9 networks (TransBTS, SwinUNETR, UNETR, TransFuse, TransUNet, UTNet) have no effective change in learned representations at the output layer when the Transformer is removed.

In combination, the above points indicate that Transformer blocks of a number of popular architectures have a minimal effect on learned representations and do not contribute to performance or even change network behavior.

## E.1   REPRESENTATIONAL SIMILARITY EXPERIMENT DETAILS

**Dataset preparation for representational similarity comparison**   Medical image segmentation methods tend to be unable to process the whole 3D volume of a single patient, instead a patch-wise approach is undertaken to predict an entire patient. Additionally, as opposed to natural images, the

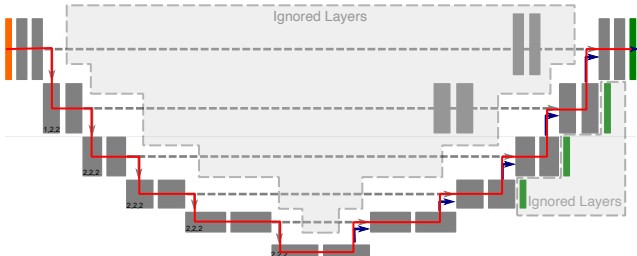

Figure 9: Visualization which positions we select to extract activations from. We select all representations at positions along the red line, after blocks that are not skipped by a residual connection.

scans usually have a fixed spacing (e.g. 1x1x1 [mm] isotropic spacing) that practitioners want to maintain. Subsequently, we use the validation cases of AMOS-CT to create a patched dataset, which we use to extract representations. Since not all architectures share an identical input patch size, we create multiple patched datasets for each input patch size, resulting in one 3D patched dataset with patches of size $96 \times 96 \times 96$ and two 2D datasets of size $224 \times 224$ and one of size $512 \times 512$. To not be subject to random augmentations, we turn off data augmentation used during training, leaving us with a preprocessed region, randomly cropped from the validation case. For each case we extract 5 patches in the 3D case and 25 patches in the 2d case, resulting in patched dataset sizes of 250 for the 3D case and 1250 for the 2D case.

**Representation extraction and comparison**   Given these patches we extract the representations of the architecture along the "outer hull" of the architecture, neglecting potential internal representation changes, to end up with a sequential-like structure (see Fig. 9). Additionally, we choose to not extract representations when residual connections are present, hence we extract either after a full Transformer block or a full CNN residual block.

**CKA calculation**   Having determined the positions to measure representations and the patched dataset to use for representation extraction, we calculate our mini-batch CKA according to Eq. (1) and Eq. (2). As for batch size, we chose 64 for all nine architectures. As we have 3 different models for all the experiments we ran, we compare all permutations of the original models to each other, resulting in three similarity values for our baseline similarity ( *'Original to Original'* values). Given the additional 3 models with their Transformer blocks replaced, we compare all 9 combinations of 1 original and 1 replaced model (*'Original to WB identity'*).

$$\text{CKA}_{minibatch}(\mathbf{K}, \mathbf{L}) = \frac{\frac{1}{k}\sum_{i=1}^{k} HSIC(K_i, L_i)}{\sqrt{\frac{1}{k}\sum_{i=1}^{k} HSIC(K_i, K_i)}\sqrt{\frac{1}{k}\sum_{i=1}^{k} HSIC(L_i, L_i)}} \qquad (1)$$

$$\text{HSIC}(\mathbf{K}, \mathbf{L}) = \frac{1}{n(n-3)}\left(tr(\tilde{\mathbf{K}}\tilde{\mathbf{L}}) + \frac{\mathbf{1}^T\tilde{\mathbf{K}}\mathbf{1}\mathbf{1}^T\tilde{\mathbf{L}}\mathbf{1}}{(n-1)(n-2)} - \frac{2}{n-2}\mathbf{1}^T\tilde{\mathbf{K}}\tilde{\mathbf{L}}\mathbf{1}\right) \qquad (2)$$

with $\mathbf{L}_i = \mathbf{X}_i\mathbf{X}_i^T$ and $\mathbf{K}_i = \mathbf{Y}_i\mathbf{Y}_i^T$ being composed of the activations of a mini-batch $\mathbf{X}_i \in \mathcal{R}^{n \times p_x}$ and $\mathbf{Y}_i \in \mathcal{R}^{n \times p_y}$. In our experiments, $p_{x/y}$ is shaped either spatially with channel, width, height, and depth dimensions or has a sequence shape of heads, tokens, and depth, which is flattened for comparison. While CKA would allow us to compare all layers in an architecture to all other layers of an architecture, we choose to only compare layers of the same index to each other, as we care about the relative change of representational similarity at these particular layers given our intervention of replacing Transformer blocks with identity mappings.

It may be important to note that all models were trained on the full 250 AMOS training cases, so there was 100% overlap between the training data of all models, with and without replacement.

**Q1: Why do we want a decreasing representational slope?**   We care about whether the Transformer blocks within the architecture contribute meaningfully to the remaining parts of the architecture. Hence we would like the Transformer blocks to change the representations as much as

Table 15: **Decision windows of clinical tasks compared to prediction times of PrimusV2 encoders.** Inference times of our encoders comfortably fit within clinical decision windows, indicating the applicability of our encoders to even the most time-critical clinical tasks. *Acq. Volume: Exemplary acquisition volume of a certain clinical task; $N_{fp}$: Number of forward passes for respective acquisition volume; $T_{pred}$: Total prediction time of the architecture.*

| Clinical Task | Decision Window | Acq. Volume | $N_{\mathbf{fp}}$ @ $160^3$ | PrimusV2 $T_{\text{pred}}$ S | B | M | L |
|---|---|---|---|---|---|---|---|
| Stroke detection (CTA) | 13 min. | [512, 512, 350] | 48 | 8.7s | 17.4s | 17.6s | 48.9s |
| Chest CT | 1-4 h | [512, 512, 500] | 64 | 11.6s | 23.2 | 23.5 | 65.2s |
| Radiotherapy Planning | 1-5 days | [512, 512, 1000] | 112 | 20.3s | 40.6s | 41.1s | 114.1s |
| Oncology Staging | Days | [512, 512, 800] | 80 | 14.5s | 29.0s | 29.4s | 81.52s |

possible from the state they had before the block. When we replace the Transformer block with an identity mapping we guarantee that current representations remain static along the block and no representational change can occur.

Given our representational comparison setting between the original architecture (starring Transformer blocks that can change the representations) and the WB identity architecture (with Transformer blocks that have been replaced with an identity mapping), we want to see that the learned Transformer blocks do something different than an identity mapping.

Should the original architectures underutilize their Transformer blocks no change occurs in them, resembling an identity mapping without being constrained to one. This will express itself in the representational similarity staying largely similar for the stretch of the Transformer blocks.

On the other hand, if the architectures utilize their Transformer blocks heavily, it will change the representations a lot, leading to a decrease in similarity to the static baseline with its Transformer blocks replaced by identity mappings.

**Q2: Why is a gap at the output desirable?** When looking at the output similarity we can interpret it as the similarity between the features used for the prediction. Given that this gap is low, we conclude that the learned features are fairly similar, while larger gaps represent less similar features.

Under this light, having replaced the Transformer block with identity mappings and observing no or a small gap indicates that the final features of the architecture without Transformers converged to a similar solution as with Transformers, indicating that the same representations can be learned by convolutions alone. On the other hand, observing a large gap indicates that the solutions the architecture with and without Transformers converges to are very different, showing that the features are changed in a way the remaining blocks are not able to achieve by themselves.

We argue that this gap indicates a good use of Transformer blocks, as it adds additional possibilities on how to solve the task, superseding what convolutions can provide by themselves. The low or no gap case instead indicates that the convolutional network can learn the same mapping as the Transformer, so why bother with the high memory demand, and more difficult training in a lower data regime, where it is not outperforming convolutions yet?

## F  CLINICAL APPLICABILITY AND INFERENCE TIMES

To assess the clinical applicability of our PrimusV2 encoders, we analyze representative 3D imaging tasks that differ in acquisition volume and clinically acceptable decision windows. These tasks span acute settings (e.g., stroke CTA), subacute diagnostics (chest CT), and longer-horizon planning workflows (radiotherapy and oncology staging). Clinical effectiveness is determined by calculating the total amount of required forward passes $N_{\text{fp}} = \left\lceil \frac{X}{160} \right\rceil \cdot \left\lceil \frac{Y}{160} \right\rceil \cdot \left\lceil \frac{Z}{160} \right\rceil$, time per forward pass $t_{\text{fp}}$ yielding the total time of prediction $T_{\text{pred}} = N_{\text{fp}} \cdot t_{\text{fp}}$. Should $T_{\text{pred}}$ be smaller than the operational time frame between image acquisition and the point at which a diagnostic or therapeutic decision must be made, we consider the model applicable in such a clinical setting.

To calculate inference times $t_{\text{fp}}$, we create a random input crop, removing the optimisation of the preprocessing pipeline, which was shown to account for a majority of the overall required prediction

time in practice (Brugnara et al., 2023). Inference times are estimated by calculating the median inference time of 100 forward passes on a single A100 gpu with a single crop per forward pass. While this is a very naive way of conducting inference, as doing many sequential forward passes of batch size one will be substantially slower than batching multiple crops together into one joint forward pass, it can serve as an upper-bound of inference time. As decision window times, we leverage values of Wong et al. (2017) for stroke detection, and estimate the remaining values, due to them being hardly reported given their non-time-critical nature.

The results of this are visualised in Table 15 and indicate that all our encoder are capable of providing prediction in time for even the most time-critical clinical tasks, despite using a naive forward pass pipeline. Our largest and slowest model, PrimusV2-L, takes a total of 48.9s for all 48 prediction required for a common acquisition window for CT-Angiography, and stays well below the maximum of 13 minutes that are prevalent in CTA image to report, leaving headroom for manual interpretation and correction of model outputs. Lastly, we denote that while all our models are theoretically applicable to time-critical tasks, it is very common to opt for faster inference at the cost of lower performance, as is the case for real-time object detection pipelines in the natural imaging domain (Robinson et al., 2025).

## G    THE USE OF LARGE LANGUAGE MODELS

Large language models (LLMs) were employed solely to improve the clarity and readability of the manuscript. They were not involved in the conception of the study, the design or execution of experiments, the analysis or interpretation of results, or any other scientific aspect of this work.

