# OpenReview forum: "Primus: Enforcing Attention Usage for 3D Medical Image Segmentation"
_ICLR.cc/2026/Conference — Submitted to ICLR 2026_

### Official Review · Reviewer_Dpm7 · 2025-10-24

**Soundness:** 2
**Presentation:** 3
**Contribution:** 2
**Rating:** 4
**Confidence:** 3

**Summary:**

The authors investigate the performance dependency of convolutional networks and Transformers in 3D segmentation tasks. They demonstrate that CNN–Transformer hybrid architectures rely heavily on the convolutional component for achieving strong segmentation performance, while the Transformer’s contribution remains minimal. To address this imbalance, the authors propose two variants of Primus that incorporate recent Transformer design advances—such as Rotary Positional Embeddings (RoPE) and EVA-style MLPs—to reduce reliance on CNNs and enhance the Transformer’s contribution in 3D segmentation.

**Strengths:**

- The paper presents a investigation comparing CNN and Transformer architectures for 3D segmentation.

- It provides many ablation studies on architectural design choices, offering valuable insights that can guide future research on Transformer-based 3D segmentation.

**Weaknesses:**

- Unclear performance gap explanation: the authors investigate the issue that CNN–Transformer hybrid architectures heavily rely on the CNN module for 3D segmentation. However, it remains unclear why CNNs consistently outperform Transformers in this domain, or how the proposed design choices specifically address this performance gap. A more detailed analysis or visualization explaining the inductive bias difference would strengthen the argument.

- Misleading problem statement: the paper argues that future 3D segmentation models should move toward pure Transformer architectures, citing advantages in self-supervised learning and multi-modal integration. However, this justification is not fully convincing. Hybrid CNN–Transformer architectures (e.g., KL-VAE, VQ-VAE, and their variants) have demonstrated strong self-supervised learning capabilities, and current research trends do not exclusively favor pure Transformers for such tasks. Moreover, hybrid models can effectively perform multimodal fusion through either Transformer or CNN blocks, provided feature dimensionalities are compatible. Thus, the necessity of abandoning CNN-based architectures remains questionable.

- High parameter count and computational cost: the proposed model family includes four sizes—Small, Base, Medium, and Large. While the Large variant achieves competitive results with nnUnet, it requires around 300M parameters and 560B FLOPs, significantly higher than other more lightweight architectures. Although the paper introduces smaller variants (Small and Base), their performance gains over prior work appear marginal, suggesting that improvements may primarily stem from scaling up model size rather than architectural innovation.

- Outdated baseline comparisons: the baseline comparisons in Table 6 seem outdated. Given the field's rapid pace, works from 2023-24 would also be acceptable. However, most of the comparison baselines are from 2021.

**Questions:**

- Pleas see weakness section.

---

> ### Author Response · Authors · 2025-11-25
> **Response Part (1/4)**
>
> > Unclear performance gap explanation: the authors investigate the issue that CNN–Transformer hybrid architectures heavily rely on the CNN module for 3D segmentation. However, it remains unclear why CNNs consistently outperform Transformers in this domain, or how the proposed design choices specifically address this performance gap. A more detailed analysis or visualization explaining the inductive bias difference would strengthen the argument.
>
> In our paper the goal is to understand _'What is holding the Transformer back?'_ and to overcome those roadblocks to build a stronger transformer architecture. In pursuit of this goal, we identify various important aspects, such as the importance of _positional encoding_, _smaller tokens_ and _insufficiencies in patch embedding_ to name a few. Additionally, we also investigated aspects that were believed to hold transformers back to be false, e.g. _the training dataset size being too small_. We highlight our empirical investigation into numerous aspects of “how to improve Transformers” in this direction:
>
> 1. **High resolution tokens:** Our experimentation (see row 1 and 2 in excerpt below from Table 3) demonstrates that the default choice of 16x16x16 tokens is ill suited for fine-grained medical structures (eg. small brain metastases lesions in SBM which are less than $0.05cm^3$) and that high resolution tokenization with 8x8x8 patches are more effective. Granular details are something CNNs with smaller kernels are suited for and our high-resolution tokens basically moves Transformers in this direction.
> 2. **Better Positional Embeddings:** We highlight that our usage of RoPE embeddings massively improves our performance (row 2 and 3 below) highlighting the default sinusoidal position embeddings commonly used in 3D medical image segmentation to be a bottleneck. While convolutions impose a structural prior that captures spatial locality and translation patterns, RoPE can be seen to introduce a strong structured representation of relative position directly into the attention mechanism.
> 3. **Strong CNN encoder:** We also introduce a stronger iterative CNN tokenizer on top of the high performing PrimusV1 Transformer to obtain our final PrimusV2 (row 4 and 5 below). The tokenizer injects stronger local inductive biases into the tokens, thereby improving our performance to a level comparable to strong CNN baselines.
>
>
> Method | 8x8x8 token patches | RoPE Embeddings |CNN tokenizer | ACDC | AMOS22 | KiTS23 | LiTS | AVG
> |----- | ------------------- | --------------- | ------------ | ---- | ----- | ----- | ---- | --- |
> ViT | No | No |No | 90.74 | 78.84 | 66.78| 71.68 | 77.01
> ViT | Yes | No | No | 91.70 | 79.81| 70.74 | 75.18 | 79.36
> ViT | Yes | Yes | No | 92.36 | 87.55 | 86.90 | 82.53 | 87.34
> PrimusV1 | Yes | Yes | No | 92.74 | 88.48 | 87.74 | 82.42 | 87.85
> PrimusV2 | Yes | Yes | Yes | 2.81 | 89.34 | 88.57 | 84.91 | 88.91
>
> While we agree that the question “Why CNNs consistently outperform Transformers” is very interesting, we look at it in reverse and try to remove obvious disadvantages that Transformers have via careful architecture design particularly in small sparsely annotated medical image segmentation datasets. We believe that our insights will be valuable to the community and hope that the reviewer shares our opinion.

---

> ### Author Response · Authors · 2025-11-25
> **Response Part (2/4)**
>
> > Misleading problem statement: the paper argues that future 3D segmentation models should move toward pure Transformer architectures, citing advantages in self-supervised learning and multi-modal integration. However, this justification is not fully convincing. Hybrid CNN–Transformer architectures (e.g., KL-VAE, VQ-VAE, and their variants) have demonstrated strong self-supervised learning capabilities, and current research trends do not exclusively favor pure Transformers for such tasks. Moreover, hybrid models can effectively perform multimodal fusion through either Transformer or CNN blocks, provided feature dimensionalities are compatible. Thus, the necessity of abandoning CNN-based architectures remains questionable.
>
> We acknowledge the confusion and definitely do not seek to dismiss hybrid CNN-Transformer architectures. Our goal is in fact focused on improving the representation learning capability of the Transformer itself regardless of its architecture context. Accordingly, we have added a new paragraph titled **CNN-Transformer Hybrid Architectures** in _Section 5, Results and Discussion_ to clarify this aspect of our work. Having clarified that, we express confusion with the reviewer’s comments as KL-VAE and VQ-VAEs are not an architecture but a training paradigm. However, regarding the broader point of self-supervised learning (SSL), it is evidenced in literature, from natural imaging, that the focus of self-supervised method development overwhelmingly favors pure transformer architectures. Since 2021, DINO [1], iBOT [2], MAE [7], DinoV2 [3], I-Jepa [4], V-Jepa [5] and DinoV3 [6] all propose their self-supervised learning methods on Vision Transformers _exclusively_. This is not the case for the medical domain and we believe this is only due to the lack of a strong transformer architecture which we provide with PrimusV1 and as PrimusV2. Through them we hope to move the 3D SSL domain towards parity with natural images to leverage SSL advancements.
>
> Regarding multi-modal integration we agree that the _output embeddings_ of CNNs and transformers both can be integrated similarly. However, the integration of multiple modalities within the architecture is substantially more organic with Transformers. E.g. including the patient-history or blood-test results into the training process can be handled more easily with Transformers. This has also become more important in the natural domain with e.g. Chameleon [8] integrating text and image tokens together. However, such approaches would be inhibited by performance bottlenecks of standard ViTs in 3D medical image analysis, requiring advancements in Transformers as in our Primus architectures.
>
> In conclusion, we are confident that Primus and Primus-V2 advances the capabilities of 3D Transformers for medical image analysis substantially and enables them for better SSL and multimodal applications. We hope we were able to convince the reviewer of the same.

---

> > ### Author Response · Authors · 2025-11-25
> > **Response Part (3/4)**
> >
> > > High parameter count and computational cost: the proposed model family includes four sizes—Small, Base, Medium, and Large. While the Large variant achieves competitive results with nnUnet, it requires around 300M parameters and 560B FLOPs, significantly higher than other more lightweight architectures. Although the paper introduces smaller variants (Small and Base), their performance gains over prior work appear marginal, suggesting that improvements may primarily stem from scaling up model size rather than architectural innovation.
> >
> > We appreciate the reviewer’s perspective; however, we believe the current evidence in our manuscript supports a different interpretation:
> > Model scaling is explicitly addressed in the paper. As noted in the manuscript, PrimusV2-Small contains only 24.6M parameters yet exceeds the default nnU-Net on 6 out of 9 tasks, with an average improvement of roughly 0.5 DSC.
> > PrimusV2-Small also substantially outperforms prior Transformer models. In particular, it surpasses CoTr on 8 out of 9 datasets, with a mean gain of 2 DSC points. Such improvements indeed go beyond being “marginal” and indicate that the architectural contributions, not parameter count, primarily drive performance.
> > Scaling trends further support this interpretation. PrimusV2-Medium (146.6M parameters) provides only a modest additional increase of 0.2 DSC on average compared to PrimusV2-Small. As discussed in L473-482 (originally L450-458), scaling benefits appear on AMOS22 and LiTS only up to the PrimusV2-M size. Increasing to PrimusV2-Large does not improve results further, underscoring that performance gains are not simply due to scaling.
> >
> > We encourage consulting the additional nuance in our discussion section on Line L473-482 in Model Scaling vs. Small Medical Datasets, which we believe is due to PrimusV2-L not converging fully. Moreover, due to low dataset scale, we believe it is difficult to show the benefits of large Models in the domain.

---

> ### Author Response · Authors · 2025-11-25
> **Response Part (4/4) and References**
>
> > Outdated baseline comparisons: the baseline comparisons in Table 6 seem outdated. Given the field's rapid pace, works from 2023-24 would also be acceptable. However, most of the comparison baselines are from 2021.
>
> We appreciate the reviewer for raising an important point in terms of newer baselines to strengthen our claims. Accordingly, we have significantly expanded our evaluation of state-of-the-art architectures in our main results table, which include the additional requested baselines, by adding MedNeXt (MICCAI-2023), SwinUNETR-V2 (MICCAI 2024), UNETR++ (IEEE Transactions on Medical Imaging 2024), SegMamba (MICCAI 2024) and VisionxLSTM (Neural Networks, Elsevier, 2024) into Table 6 in the manuscript. We train these models using the same rigorous settings as in our pre-existing baselines, over all 9 datasets - resulting in training 225 additional individual models over X GPU Hours worth of compute.
>
> Methods | ACDC | AMOS22 | KiTS23 | LiTS | SST3 | MAMA | SBM | Atlas22 | Word | Avg
> |------|-------|--------|--------|------|------|------|-----|---------|-----|------|
> _CNN Baselines_
> nnUNet                 | 91.34 | 88.61 | 85.99 | 79.29 | **90.27** | 78.32 | **66.52** | 63.11 | 83.11 | 80.73
> ResEncL              | 92.54 | **89.39** | **88.06** | **81.20** | **90.28** | **79.00** | 64.00 | **63.12** | **85.79** | **81.48**
> MedNeXt-L (new) | **92.55** | **89.58** | **88.20** | **81.57** | **89.93** | **79.42** | **65.85** | **63.03** | **85.37** | **81.72**
> _Other Baselines_
> SegMamba  (new) | **92.65** | 86.60 | 82.52 | 76.89 | 89.19 | 75.67 | 63.58 | 62.23 | 82.95 | 79.14 |
> VisionxLSTM (new) | 92.41 | 78.30 | 80.49 | 74.61 | 88.32 | 73.54 | 61.04 | 60.09 | 74.85 | 75.96
> _Transformer Baselines_
> CoTR                   | 90.50 | 87.93 | 84.63 | 78.44 | 89.60 | 76.95 | 59.96 | 62.14 | 83.11 | 79.25
> nnFormer             | 92.35 | 81.35 | 75.72 | 77.02 | 88.62 | 68.71 | 64.37 | 60.61 | 82.53 | 76.81
> SwinUNETR        | 91.11 | 80.88 | 75.07 | 73.27 | 88.60 | 75.55 | 61.96 | 60.56 | 78.99 | 76.22
> UNETR                | 90.41 | 62.93 | 76.33 | 70.91 | 86.73 | 74.27 | 49.87 | 54.15 | 70.87 | 70.72
> UNETR++ (new)  | **92.58** | 84.82 | 82.42 | 77.69 | 89.50 | 77.16 | 64.33 | 61.39 | 77.72 | 78.62
> SwinUNETR-v2 (new) | 91.94 | 86.17 | 83.99 | 77.18 | 88.72 | 75.84 | 60.80 | 61.10 | 82.20 | 78.66
> _Our Models (only M for brevity)_
> Primus-M             | 91.82 | 87.47 | 85.87 | 78.97 | 87.83 | 77.14 | 56.71 | 60.21 | 82.84 | 78.76
> PrimusV2-M         | 92.27 | **89.35** | **88.09** | **81.73** | 88.26 | **79.40** | **66.36** | **63.23** | **84.15** | **81.43**
>
> We have highlighted the top 3 values in each column in Markdown. It can be observed that on the datasets evaluated, the state-of-the-art CNNs of ResEncL and MedNeXt-L and our PrimusV2-M trade top-3 performances across all 9 datasets. We see other Transformer baselines as well as Mamba or xLSTM baselines trail significantly in performance. Importantly, PrimusV2 is the _only_ Transformer-based architecture to be within 0.05 DSC and 0.29 DSC of state-of-the-art CNNs such as ResEnc-L and MedNeXt respectively in overall performance. In doing so, we strengthen our position as one of the first attempts at effectively leveraging the Transformer architecture for learning strong representation for 3D medical image segmentation. We fully expect our work to open new avenues for method development in self-supervised learning and multi-modal techniques. We hope that our experimentation was able to address the issues raised in the review.
>
> --------
>
> ### References
>
> [1] Caron, Mathilde, et al. "Emerging properties in self-supervised vision transformers." Proceedings of the IEEE/CVF international conference on computer vision. 2021.
>
> [2] Zhou, Jinghao, et al. "ibot: Image bert pre-training with online tokenizer." arXiv preprint arXiv:2111.07832 (2021).
>
> [3] Oquab, Maxime, et al. "Dinov2: Learning robust visual features without supervision." arXiv preprint arXiv:2304.07193 (2023).
>
> [4] Assran, Mahmoud, et al. "Self-supervised learning from images with a joint-embedding predictive architecture." Proceedings of the IEEE/CVF Conference on Computer Vision and Pattern Recognition. 2023.
>
> [5] Bardes, Adrien, et al. "V-jepa: Latent video prediction for visual representation learning." (2023).
>
> [6] Siméoni, Oriane, et al. "Dinov3." arXiv preprint arXiv:2508.10104 (2025).
>
> [7] He, Kaiming, et al. "Masked autoencoders are scalable vision learners." Proceedings of the IEEE/CVF conference on computer vision and pattern recognition. 2022.
>
> [8] Team, Chameleon. "Chameleon: Mixed-modal early-fusion foundation models." arXiv preprint arXiv:2405.09818 (2024).

---

### Official Review · Reviewer_n3eQ · 2025-10-31

**Soundness:** 2
**Presentation:** 2
**Contribution:** 2
**Rating:** 4
**Confidence:** 3

**Summary:**

This paper presents PRIMUS and PRIMUS-V2, two Transformer-centric architectures for 3D medical image segmentation. The authors argue that previous hybrid CNN–Transformer models rely heavily on convolutional components and fail to demonstrate the effectiveness of Transformer blocks. PRIMUS is designed to “enforce attention usage” through architectural changes such as high-resolution tokens, 3D rotary position embeddings, and lightweight decoders. Experiments across multiple datasets show that PRIMUS and PRIMUS-V2 achieve performance comparable to or slightly exceeding strong CNN baselines such as nnU-Net and ResEnc-L.

**Strengths:**

1. Comprehensive empirical analysis: The paper provides an extensive evaluation across nine public datasets and multiple baselines, demonstrating strong experimental rigor.

2. Clear architecture design and ablation studies: The paper systematically explores the effects of each architectural modification, such as patch size, position embedding, and tokenizer design.

3. Good reproducibility and transparency: The authors provide detailed training configurations, public datasets, and plan to release code integrated with nnU-Net, which increases reproducibility.

**Weaknesses:**

1. Questionable motivation: The motivation is somewhat inconsistent. In the introduction (lines 54–68), the authors acknowledge that CNNs currently outperform Transformers in 3D medical segmentation. This raises the question of why a Transformer-centric model is necessary. Furthermore, the stated advantages of Transformers (e.g., efficiency in self-supervised learning and multimodal integration) could arguably also be achieved with hybrid CNN–Transformer models, which are dismissed too quickly. The motivation therefore needs substantial clarification and rewriting.

2. Outdated baselines: Most of the compared methods are from 2024 or earlier. Considering that this paper targets ICLR 2026, comparisons to more recent 2025 Transformer or CNN-based segmentation models are needed to ensure fairness and relevance.

3. Limited performance gain: The proposed method achieves only marginal improvements (see Table 6), and sometimes even lags behind strong CNNs such as ResEnc-L. This weak advantage raises concerns about whether it is truly necessary to pursue a Transformer-centric architecture for 3D medical segmentation.

**Questions:**

Please see the weaknesses.

---

> ### Author Response · Authors · 2025-11-25
> **Response Part (1/3)**
>
> > Questionable motivation: The motivation is somewhat inconsistent. In the introduction (lines 54–68), the authors acknowledge that CNNs currently outperform Transformers in 3D medical segmentation. This raises the question of why a Transformer-centric model is necessary. Furthermore, the stated advantages of Transformers (e.g., efficiency in self-supervised learning and multimodal integration) could arguably also be achieved with hybrid CNN–Transformer models, which are dismissed too quickly. The motivation therefore needs substantial clarification and rewriting.
>
> In recent years, the natural image domain has been dominated by pure transformer architectures. At the same time, transformers have failed to make a meaningful entry into the 3D medical imaging domain, with competitions still exclusively being won by CNN designs. This seems quite surprising given that there is no tell why such a clear performance benefit from natural images would not translate to 3D images. In itself, this already warrants a scientific investigation such as the one done in our paper. But let’s take a step further and think about what this means for progress: Large parts of the natural imaging community focus on squeezing more performance and capabilities out of the transformer architectures used in this domain. Self-supervised learning in particular has made substantial progress, with DINO [1], iBOT [3], MAE [7], DinoV2 [2], I-Jepa [4], V-Jepa [5] and DinoV3 [6] proposing groundbreaking advancements in pretraining task formulation that unlock new levels of performance from training on unlabeled data: a key capability highly sought after also in the medical domain but _only formulated for pure transformer architectures_. While CNN adaptations of these objectives may to some extent exist or be in the process of development, transfer of new paradigms between the medical and natural imaging domain is severely hampered by the fact that both rely on different classes of architectures to achieve state of the art performance. The proposed PrimusV1 and V2 architectures offer a direct path for translating insights between domains, promising faster adaptation of new ideas without sacrificing performance on 3D data.
>
> Regarding multi-modal integration we agree that the _output embeddings_ of CNNs and transformers both can be integrated similarly. However, the integration of multiple modalities within the architecture is substantially more organic with Transformers. E.g. including the patient-history or blood-test results into the training process can be handled more easily with Transformers. This has also become more important in the natural domain with e.g. Chameleon [9] integrating text and image tokens together. However, such approaches would be inhibited by performance bottlenecks of standard ViTs in 3D medical image analysis, requiring advancements in Transformers as in our Primus architectures.
>
> In conclusion, we definitely do not seek to dismiss hybrid CNN-Transformer architectures. Our goal is in fact focused on improving the representation learning capability of the Transformer itself. Accordingly, we have added a new paragraph titled **CNN-Transformer Hybrid Architectures** in _Section 5, Results and Discussion_ to clarify this aspect of our work.
>
> To avoid confusion of future readers of our paper, **we extend this clarification in the Introduction in the “Why Transformers?”** paragraph as requested by the reviewer. We are deeply thankful to the reviewer for giving us an opportunity to address this aspect and improve our paper.

---

> ### Author Response · Authors · 2025-11-25
> **Reponse Part (2/3)**
>
> > Outdated baselines: Most of the compared methods are from 2024 or earlier. Considering that this paper targets ICLR 2026, comparisons to more recent 2025 Transformer or CNN-based segmentation models are needed to ensure fairness and relevance.
>
> We thank the reviewer for raising such an important point in terms of newer baselines to strengthen our claims. In order to address this, we have significantly expanded our evaluation of newer state-of-the-art architectures in our main results table, by adding MedNeXt (MICCAI-2023), SwinUNETR-V2 (MICCAI 2024), UNETR++ (IEEE Transactions on Medical Imaging 2024), SegMamba (MICCAI 2024) and VisionxLSTM (Neural Networks, Elsevier, 2024) into Table 6 in the manuscript. We train these models using the same rigorous settings as in our pre-existing baselines, over all 9 datasets - resulting in training 225 additional individual models to further strengthen our results.
>
> Methods | ACDC | AMOS22 | KiTS23 | LiTS | SST3 | MAMA | SBM | Atlas22 | Word | Avg
> |------|-------|--------|--------|------|------|------|-----|---------|-----|------|
> CNN Baselines
> nnUNet                 | 91.34 | 88.61 | 85.99 | 79.29 | **90.27** | 78.32 | **66.52** | 63.11 | 83.11 | 80.73
> ResEncL              | 92.54 | **89.39** | **88.06** | **81.20** | **90.28** | **79.00** | 64.00 | **63.12** | **85.79** | **81.48**
> MedNeXt-L (new) | **92.55** | **89.58** | **88.20** | **81.57** | **89.93** | **79.42** | **65.85** | **63.03** | **85.37** | **81.72**
> Other Baselines
> SegMamba  (new) | **92.65** | 86.60 | 82.52 | 76.89 | 89.19 | 75.67 | 63.58 | 62.23 | 82.95 | 79.14 |
> VisionxLSTM (new) | 92.41 | 78.30 | 80.49 | 74.61 | 88.32 | 73.54 | 61.04 | 60.09 | 74.85 | 75.96
> Transformers
> CoTR                   | 90.50 | 87.93 | 84.63 | 78.44 | 89.60 | 76.95 | 59.96 | 62.14 | 83.11 | 79.25
> nnFormer             | 92.35 | 81.35 | 75.72 | 77.02 | 88.62 | 68.71 | 64.37 | 60.61 | 82.53 | 76.81
> SwinUNETR        | 91.11 | 80.88 | 75.07 | 73.27 | 88.60 | 75.55 | 61.96 | 60.56 | 78.99 | 76.22
> UNETR                | 90.41 | 62.93 | 76.33 | 70.91 | 86.73 | 74.27 | 49.87 | 54.15 | 70.87 | 70.72
> UNETR++ (new)  | **92.58** | 84.82 | 82.42 | 77.69 | 89.50 | 77.16 | 64.33 | 61.39 | 77.72 | 78.62
> SwinUNETR-v2 (new) | 91.94 | 86.17 | 83.99 | 77.18 | 88.72 | 75.84 | 60.80 | 61.10 | 82.20 | 78.66
> Our Models _(only M for brevity in Markdown)_
> Primus-M             | 91.82 | 87.47 | 85.87 | 78.97 | 87.83 | 77.14 | 56.71 | 60.21 | 82.84 | 78.76
> PrimusV2-M         | 92.27 | **89.35** | **88.09** | **81.73** | 88.26 | **79.40** | **66.36** | **63.23** | **84.15** | **81.43**
>
> We have highlighted the top 3 values in each column in Markdown. It can be observed that on the datasets evaluated, the state-of-the-art CNNs of ResEncL and MedNeXt-L and our PrimusV2-M trade top-3 performances across the majority of datasets. We see other Transformer baselines as well as Mamba or xLSTM baselines trail significantly in performance. Importantly, PrimusV2 is the _only_ Transformer-based architecture to be within 0.05 DSC and 0.29 DSC of state-of-the-art CNNs such as ResEnc-L and MedNeXt respectively in overall performance. In doing so, we strengthen our position as one of the first attempts at effectively leveraging the Transformer architecture for learning strong representation for 3D medical image segmentation. We fully expect our work to open new avenues for method development in self-supervised learning and multi-modal techniques. We hope that our experimentation was able to address the issues raised in the review. We look forward to your reply and a positive evaluation.

---

> > ### Author Response · Authors · 2025-11-25
> > **Response Part (3/3) and References**
> >
> > > Limited performance gain: The proposed method achieves only marginal improvements (see Table 6), and sometimes even lags behind strong CNNs such as ResEnc-L. This weak advantage raises concerns about whether it is truly necessary to pursue a Transformer-centric architecture for 3D medical segmentation.
> >
> > We appreciate the reviewer’s perspective and would like to offer two clarifications:
> >
> > 1. Transformers are the prevalent architectures used in natural imaging SSL research, with the latest methods all using ViTs for self-supervised learning [1-7]. By providing a transformer that is on-par with 3D CNNs we can unlock the transfer of new paradigms from natural imaging, such as SSL methods, to 3D medical imaging quickly.
> >     1. Methods don’t need to be adapted from ViTs to CNNs anymore
> >     2. Compute efficiency can be increased by dropping masked tokens.
> >     3. Transformers don’t have a performance gap they need to close with their pre-training to be “on-par” with CNNs, as was the case in the OpenMind benchmark [8].
> > 2. While PrimusV2 admittedly does not clearly surpass ResEnc-L and MedNeXt, the stated goal of our line of research is to push the boundaries of transformer-based architectures for 3D medical image segmentation. As such, the baseline Primus should be compared to the current transformer hybrid architectures, which all fall well short of both Primus itself and the strong CNN baselines. Notably, PrimusV2 improves performance by 2 DSC points relative to the next best contender (CoTr). We see our work as a meaningful first step towards unlocking the potential of transformers for 3D, and have already identified several key aspects that must be considered in 3D such as the role of RoPE. We hope our work inspires more research in this direction.
> >
> > Combining these two points, we are convinced there is a need for a transformer-architecture in 3D medical segmentation, which PrimusV2 fills, even if it is “just” on-par with current state-of-the-art CNNs.
> >
> > ------
> >
> > ### References:
> > [1] Caron, Mathilde, et al. "Emerging properties in self-supervised vision transformers." Proceedings of the IEEE/CVF international conference on computer vision. 2021.
> >
> > [2] Oquab, Maxime, et al. "Dinov2: Learning robust visual features without supervision." arXiv preprint arXiv:2304.07193 (2023).
> >
> > [3] Zhou, Jinghao, et al. "ibot: Image bert pre-training with online tokenizer." arXiv preprint arXiv:2111.07832 (2021).
> >
> > [4] Assran, Mahmoud, et al. "Self-supervised learning from images with a joint-embedding predictive architecture." Proceedings of the IEEE/CVF Conference on Computer Vision and Pattern Recognition. 2023.
> >
> > [5] Bardes, Adrien, et al. "V-jepa: Latent video prediction for visual representation learning." (2023).
> >
> > [6] Siméoni, Oriane, et al. "Dinov3." arXiv preprint arXiv:2508.10104 (2025).
> >
> > [7] He, Kaiming, et al. "Masked autoencoders are scalable vision learners." Proceedings of the IEEE/CVF conference on computer vision and pattern recognition. 2022.
> >
> > [8] Wald, Tassilo, et al. "An OpenMind for 3D medical vision self-supervised learning." Proceedings of the IEEE/CVF International Conference on Computer Vision. 2025.
> >
> > [9] Team, Chameleon. "Chameleon: Mixed-modal early-fusion foundation models." arXiv preprint arXiv:2405.09818 (2024).

---

### Official Review · Reviewer_J9TG · 2025-10-31

**Soundness:** 2
**Presentation:** 3
**Contribution:** 3
**Rating:** 6
**Confidence:** 2

**Summary:**

This paper proposes Primus and PrimusV2, two transformer‑centric architectures for 3D medical image segmentation, and a diagnostic study that quantifies how much current transformer models actually depend on attention. The paper introduces a UNet Index to measure non‑transformer capacity, shows that removing attention often barely hurts performance in popular hybrids, and then designs Primus with high‑resolution tokens, 3D Rotary Position Embeddings, Eva‑02‑style SwiGLU MLPs, LayerScale, and a lightweight decoder. PrimusV2 adds an iterative tokenizer to better capture small structures. Across nine datasets, PrimusV2 is competitive with a strong CNN (ResEnc‑L) and clearly outperforms prior Transformer baselines under a unified nnU‑Net training setup, with ablations and scaling experiments that support the design choices.

**Strengths:**

- The UNet Index plus the remove‑the‑Transformer ablation offer a clear, quantitative look at hybrid designs and motivate a truly attention‑centric approach; this analysis helps explain why many prior hybrids trail well‑tuned CNNs.
- The architecture is carefully built from principled components for the 3D setting (8×8×8 tokens, 3D RoPE, SwiGLU, LayerScale), and the authors run systematic ablations that show steady gains and improved stability.
- The evaluation spans nine datasets with consistent training in nnU‑Net, qualitative examples, and clear reporting that PrimusV2 matches or exceeds nnU‑Net on most sets and approaches ResEnc‑L on average.

**Weaknesses:**

The baseline comparison omits several recent and strong segmentation models, which makes the comparisons feel dated, in particular, there are no results against MedNeXt, UNETR++, SwinUNETR‑V2, or SegMamba on the same splits and preprocessing, so it is hard to judge progress relative to the current works.

**Questions:**

Could you clarify why comparisons to recent methods such as MedNeXt, UNETR++, SwinUNETR-V2, and SegMamba were not included, and can you compare with them under the same setting on the reported datasets?

---

> ### Author Response · Authors · 2025-11-25
> **Response Part (1/1)**
>
> > The baseline comparison omits several recent and strong segmentation models, which makes the comparisons feel dated, in particular, there are no results against MedNeXt, UNETR++, SwinUNETR‑V2, or SegMamba on the same splits and preprocessing, so it is hard to judge progress relative to the current works.
>
> We thank the reviewer for raising an important point in terms of newer baselines to strengthen our claims. In order to satisfy this requirement, we have significantly expanded our evaluation of state-of-the-art architectures in our main results table, which include the additional requested baselines, by adding MedNeXt (MICCAI-2023), SwinUNETR-V2 (MICCAI 2024), UNETR++ (IEEE Transactions on Medical Imaging 2024), SegMamba (MICCAI 2024) and VisionxLSTM (Neural Networks, Elsevier, 2024) into Table 6 in the manuscript. We train these models using the same rigorous settings as in our pre-existing baselines, over all 9 datasets - resulting in training 225 additional individual models to strengthen our results.
>
> Methods | ACDC | AMOS22 | KiTS23 | LiTS | SST3 | MAMA | SBM | Atlas22 | Word | Avg
> |------|-------|--------|--------|------|------|------|-----|---------|-----|------|
> _CNN Baselines_
> nnUNet                 | 91.34 | 88.61 | 85.99 | 79.29 | **90.27** | 78.32 | **66.52** | 63.11 | 83.11 | 80.73
> ResEncL              | 92.54 | **89.39** | **88.06** | **81.20** | **90.28** | **79.00** | 64.00 | **63.12** | **85.79** | **81.48**
> MedNeXt-L (new) | **92.55** | **89.58** | **88.20** | **81.57** | **89.93** | **79.42** | **65.85** | **63.03** | **85.37** | **81.72**
> _Other Baselines_
> SegMamba  (new) | **92.65** | 86.60 | 82.52 | 76.89 | 89.19 | 75.67 | 63.58 | 62.23 | 82.95 | 79.14 |
> VisionxLSTM (new) | 92.41 | 78.30 | 80.49 | 74.61 | 88.32 | 73.54 | 61.04 | 60.09 | 74.85 | 75.96
> _Transformer Baselines_
> CoTR                   | 90.50 | 87.93 | 84.63 | 78.44 | 89.60 | 76.95 | 59.96 | 62.14 | 83.11 | 79.25
> nnFormer             | 92.35 | 81.35 | 75.72 | 77.02 | 88.62 | 68.71 | 64.37 | 60.61 | 82.53 | 76.81
> SwinUNETR        | 91.11 | 80.88 | 75.07 | 73.27 | 88.60 | 75.55 | 61.96 | 60.56 | 78.99 | 76.22
> UNETR                | 90.41 | 62.93 | 76.33 | 70.91 | 86.73 | 74.27 | 49.87 | 54.15 | 70.87 | 70.72
> UNETR++ (new)  | **92.58** | 84.82 | 82.42 | 77.69 | 89.50 | 77.16 | 64.33 | 61.39 | 77.72 | 78.62
> SwinUNETR-v2 (new) | 91.94 | 86.17 | 83.99 | 77.18 | 88.72 | 75.84 | 60.80 | 61.10 | 82.20 | 78.66
> **Our Models** _(only M for brevity in Markdown)_
> Primus-M             | 91.82 | 87.47 | 85.87 | 78.97 | 87.83 | 77.14 | 56.71 | 60.21 | 82.84 | 78.76
> PrimusV2-M         | 92.27 | **89.35** | **88.09** | **81.73** | 88.26 | **79.40** | **66.36** | **63.23** | **84.15** | **81.43**
>
> We have highlighted the top 3 values in each column in Markdown and included them in the revised Manuscript in Table 6. It can be observed that on the datasets evaluated, the state-of-the-art CNNs of ResEncL and MedNeXt-L and our PrimusV2-M trade top-3 performances across almost all 9 datasets. We see other Transformer baselines as well as Mamba or xLSTM baselines trail significantly in performance. Importantly, PrimusV2 is the _only_ Transformer-based architecture to be within 0.05 DSC and 0.29 DSC of state-of-the-art CNNs such as ResEnc-L and MedNeXt respectively in overall performance. In doing so, we strengthen our position as one of the first attempts at effectively leveraging the Transformer architecture for learning strong representation for 3D medical image segmentation. We fully expect our work to open new avenues for method development in self-supervised learning and multi-modal techniques. We hope that our experimentation was able to address the issues raised in the review and, as stated in the abstract, fully commit to publishing our source code. We look forward to your reply and a positive evaluation.

---

### Official Review · Reviewer_Vcda · 2025-11-01

**Soundness:** 3
**Presentation:** 4
**Contribution:** 3
**Rating:** 4
**Confidence:** 3

**Summary:**

This manuscript addresses a critical gap in 3D medical image segmentation: despite the success of Transformers in other domains, state-of-the-art methods in this field remain dominated by convolutional neural networks, as existing Transformer-based models suffer from over-reliance on convolutional blocks and redundant Transformer layers.
To solve this, the authors propose two Transformer-centric architectures: Primus and PrimusV2. Key innovations include 8×8×8 small-patch tokenization for Primus, iterative patch embedding via residual blocks for PrimusV2, 3D axial Rotary Position Embedding, and a lightweight convolutional decoder. Experiments on 9 public datasets show Primus competes with default nnU-Net, while PrimusV2 matches the SOTA CNN ResEnc-L, marking the first competitive Transformer-centric model for 3D medical segmentation.

**Strengths:**

1.	Domain-Specific Architectural Innovations: Innovations are tailored to 3D medical images’ unique challenges: (1) 8×8×8 patch tokenization addresses the local detail loss from UNETR’s 16×16×16 patches (critical for small lesions like SBM’s <0.05 cm³ metastases); (2) 3D RoPE resolves spatial awareness issues of absolute position embeddings (ablations show DSC improves from 77.13 to 87.84, Table 4); (3) PrimusV2’s iterative patch embedding solves Transformer’s difficulty in learning low-level features with small data—all supported by clear ablation results (Table 3, Table 12).
2.	Comprehensive, Reproducible Experiments: The work benchmarks against 6 SOTA baselines across 9 datasets, covering multiple modalities (CT, MRI) and tasks. It also provides implementation details and plans to release code, ensuring reproducibility.

**Weaknesses:**

1.	The manuscript analyzes 9 mainstream architectures and finds that their original studies did not verify performance changes after removing Transformer blocks. However, some of these models (e.g., Transfuse, TransBTS) did conduct ablations on attention windows, position embeddings, or Transformer layer counts. Could you clarify: (1) Whether you reviewed the ablation sections of these original studies to confirm they indeed omitted "Transformer block removal" experiments? (2) For models that did test partial Transformer component adjustments (e.g., reducing attention heads), why do you think such ablations fail to reflect the redundancy of Transformer blocks as a whole?
2.	Primus-L and PrimusV2-L have 325.5M and 326.1M parameters, respectively, far exceeding the default nnU-Net’s ~30M parameters. Even with a lightweight decoder, the 8× longer token sequence (from 8×8×8 patches) increases inference latency, which is critical for clinical deployment.
3.	The core of the residual blocks in PrimusV2 lies in convolutional operations, whose role is to pre-learn low-level features, which should originally be learned by the Transformer. The ablation experiment in Appendix C.2 of the manuscript shows that for PrimusV2-M using Minimal Residual Downsampling (MRD), after removing the Transformer blocks, the average Dice Similarity Coefficient still reaches 83.98. In contrast, for Primus-M using "convolution-free Default Projection (DP)", the average DSC drops to only 19.78 after Transformer removal. This indicates that the convolutional residual blocks in PrimusV2 can independently learn a large number of effective features and maintain relatively high performance even without the Transformer, which contradicts the core logic of "Transformer-centric" architecture that requires "the Transformer to dominate representation learning".

**Questions:**

1.	Please refer to Weakness 1.
2.	Regarding computational cost: Have you compared the inference latency and memory usage of Primus/PrimusV2 with baseline models on standard clinical hardware to evaluate deployment feasibility?
3.	Please refer to Weakness 3.

---

> ### Author Response · Authors · 2025-11-25
> **Response Part (1/3)**
>
> > The manuscript analyzes 9 mainstream architectures and finds that their original studies did not verify performance changes after removing Transformer blocks. However, some of these models (e.g., Transfuse, TransBTS) did conduct ablations on attention windows, position embeddings, or Transformer layer counts. Could you clarify: (1) Whether you reviewed the ablation sections of these original studies to confirm they indeed omitted "Transformer block removal" experiments? (2) For models that did test partial Transformer component adjustments (e.g., reducing attention heads), why do you think such ablations fail to reflect the redundancy of Transformer blocks as a whole?
>
> Thank you for the insightful comment and giving us a chance to discuss this in further detail. We did indeed review the ablations of the nine cited papers. Three of them, namely, TransUNet, CoTr and TransFuse, do perform a similar transformer removal ablation in their work. However, in their work they find the transformer to be beneficial, which we believe can be attributed to the following aspects of their experiments:
> 1. __CoTR__ shows a decrease of 1.6 DSC points on removal of its Transformer. However its conclusions are drawn off of a single 9 sample split on the BTCV dataset. This dataset was shown to be highly noisy and not a great choice for benchmarking [1], which may mask the ineffectiveness of their findings.
> 2. __TransUNet__ shows a decrease of 2.8 DSC points when removing their transformer. Similarly to CoTR they evaluate their findings on a single 12 sample split on the BTCV dataset, so the same noisiness of the dataset might be the issue.
> 3. __TransFuse__ evaluates this ablation on a single fold of two datasets, KvasirSeg and ColonDB. They show 1.8 DSC points decrease on Kvasir and 12.2 DSC on ColonDB when removing their transformer. A potential reason for why the lack of Transformer performance may not be visible on their end, could be due to the evaluation datasets being 2D surgical data, which may require more global reasoning due to e.g. image occlusions prevalent in natural images that do not exist in 3D medical image segmentation.
>
> Compared to these papers, our ablations are conducted with a 3-fold cross-validation on two robust benchmarking datasets including notably a test set of 251 CT volumes in TotalSegmentator-BTCV. This is significantly more than the original Transformer papers and yields overall more reliable results. Further, we care about 3D medical segmentation datasets, which is a very different domain compared to surgical data. We also explore the effect of training set size on the performance of such Transformers ablation by varying samples from 1 - 100% out of 1000 samples on TotalSegmentator-BTCV, revealing further insights into Transformer ineffectiveness.

---

> > ### Author Response · Authors · 2025-11-25
> > **Response Part (2/3)**
> >
> > > Primus-L and PrimusV2-L have 325.5M and 326.1M parameters, respectively, far exceeding the default nnU-Net’s ~30M parameters. Even with a lightweight decoder, the 8× longer token sequence (from 8×8×8 patches) increases inference latency, which is critical for clinical deployment.
> >
> > To evaluate the possibility of our architectures for clinical deployment, we evaluated the average time of a forward pass and compared it to the time available between image acquisition and the clinical decision being made for various clinical tasks. In particular, we evaluate our slower PrimusV2-L, which takes `1.019`s per “160x160x160” input and PrimusV2-S which takes `0.1810`s per 160x160x160 input. _An extended evaluation table and discussion of this is now included in Appendix F of our revised manuscript._
> >
> >  We list a table of clinical tasks with their approximate required prediction times below:
> >
> > | Clinical Task | Decision Window | Acquisition volume in pixels | Forwards required @ 160 | Prediction Time PrimusV2-L | Prediction Time PrimusV2-S |
> > |---------------|-----------------|------------------------------|------------------------------|------------------------------|------------------------------|
> > | Stroke detection (CTA head/neck) | 13 min [3] | 512 × 512 × 350 | 48 | 48.9s | 8.7s |
> > | Chest CT | 1–4 hours | 512 × 512 × 500 | 64 | 65.2s | 11.6s |
> > | Radiotherapy Planning CT (CT-Sim) | 1–5 days | 512 × 512 × 1000 | 112 | 114.1s | 20.3s |
> > | Oncology CT (Staging) | Days | 512 × 512 × 800 | 80 | 81.52s | 14.5s |
> >
> > As can be seen, even our slowest and largest encoder is capable of being applied to the most time-critical clinical task, which is Stroke Detection in 49s total time. With the decision window being 13 minutes and other stroke AI models targeting inference in ~2 minutes [2] our inference time is well within this window and would leave plenty of time for manual correction despite running on a single A100 gpu as opposed to 4xA100s as in [2]. Most notably, this prediction time is an upper-bound of inference time, as we conduct a forward-pass with a single crop, while one could batch multiple crops together to improve throughput, or even parallelize inference across multiple GPUs to reduce the inference time further.
> >
> > More importantly we want to emphasize that most clinical tasks do not require particularly fast inference times. For example,
> > 1. Chest CTs are needed to know if a patient has pleural effusion or pneumonia.
> > 2. Radiotherapy planning requires segmented structures to determine radiation dosage.
> > 3. Oncology staging is needed to accurately assess tumor burden and guide clinical decision making.
> >
> > All of these tasks help guide clinical decision making but **do not require a decision in minutes**. Overall, we want to emphasize that the inference time of our model does not influence its clinical applicability and is still at a size and inference speed to be applicable to even the most time-critical tasks as in stroke detection.
> > For completion, we provide a dedicated section on this with an extended table in  Appendix F in our revised paper, highlighting the forward-pass times of all PrimusV2 architectures with an extended discussion about the clinical applicability of the encoders for different clinical tasks.

---

> > > ### Author Response · Authors · 2025-11-25
> > > **Response Part (3/3)**
> > >
> > > >The core of the residual blocks in PrimusV2 lies in convolutional operations, whose role is to pre-learn low-level features, which should originally be learned by the Transformer. The ablation experiment in Appendix C.2 of the manuscript shows that for PrimusV2-M using Minimal Residual Downsampling (MRD), after removing the Transformer blocks, the average Dice Similarity Coefficient still reaches 83.98. In contrast, for Primus-M using "convolution-free Default Projection (DP)", the average DSC drops to only 19.78 after Transformer removal. This indicates that the convolutional residual blocks in PrimusV2 can independently learn a large number of effective features and maintain relatively high performance even without the Transformer, which contradicts the core logic of "Transformer-centric" architecture that requires "the Transformer to dominate representation learning".
> > >
> > > We thank the reviewer for raising an important point and allowing us to clarify our stance with hybrid architectures. In fact, recontextualizing your statements actually strengthens our message instead of weakening it. We do so as follows:
> > >
> > > 1. **Hybrid architectures are not the problem:** Firstly, we do not discourage the idea of hybrid architectures. Our core claim is that “current hybrid architectures hide Transformer ineffectiveness” - not that either Transformer themselves or the principle of hybrid architectures is the problem. The goal of our research was to identify how transformers must be implemented into a segmentation algorithm in order to meaningfully contribute to its performance, rather than simply being added to an already functioning CNN for the sake of scientific novelty. PrimusV1 offers exactly that - a Transformer-centric architecture which delivers near state-of-the-art performance while relying almost exclusively on the Transformer. PrimusV2 takes deliberate steps towards hybridization by including stronger CNN components where needed. The inclusion of MRD not only pushes its performance to parity with ResEnc-L, but importantly hints at tokenization as a major bottleneck in 3D medical image segmentation with pure transformer architectures. Thus, the presented work essentially approaches hybrid architectures from the opposite direction of related work: instead of starting from CNNs and adding Transformers, we start from a proven Transformer (PrimusV1) and surgically add minimal CNN components where needed to maximize performance.
> > >
> > >
> > > 2. **PrimusV2 as a Transformer-centric network:** Comparing our purest transformer i.e. PrimusV1 with PrimusV2, we denote that PrimusV1 can be considered PrimusV2 without the CNN (the MRD). Removing the Transformer from PrimusV1 reduces performance by ~68 DSC (see row 1 and 2 of excerpt from Table 3 below) demonstrating the power of our Transformer in learning representations. In Primus V2, we essentially pair PrimusV1 with a stronger CNN tokenizer which increases performance by ~1 DSC points (row 2 and 3). However, if we were to now remove the Transformer from PrimusV2, we would lose about ~5 DSC points of accuracy. While the stronger CNN tokenizer is still able to learn better features than the V1 encoder it still requires the contribution of the strong Primus Transformer to reach anywhere near state-of-the-art performance. This clearly indicates that a proven-to-work transformer coupled with a powerful CNN tokenizer is a stronger combination than either alone. So while PrimusV2 is not as “Transformer-centric” as PrimusV1 is, it effectively constitutes the highest performing transformer-CNN hybrid architecture to date.
> > >
> > > Method | v2-CNN Tokenizer | Transformer | ACDC | AMOS22 | KiTS23 | LiTS | AVG
> > > |----- | ---------------- | ----------- | ---- | ------ | ------ | ---- | --- |
> > > Primus V1 | No | No | 19.45 | 03.02 | 15.88 | 40.76 | 19.78
> > > Primus V1 | No | Yes | 92.74 | 88.48 | 87.74 | 82.42| 87.85
> > > Primus V2 | Yes | Yes | 92.81 | 89.34 | 88.57 | 84.91| 88.91
> > > Primus V2 | Yes | No | 91.89 | 81.56 | 82.69 | 79.77 | 83.98
> > >
> > > To prevent other from having this confusion when reading the paper we have added a new paragraph titled **CNN-Transformer Hybrid Architectures** in _Section 5, Results and Discussion_ specifically addressing that we are not against hybrid architectures, and instead more interested in making a Transformer backbone contribute effectively in conjunction or absence of CNN components.
> > >
> > > In conclusion, we hope our responses have been satisfactory in addressing the issues raised by the reviewer and look forward to a positive evaluation.

---

> > > > ### Author Response · Authors · 2025-11-25
> > > > **Additional Remarks and References**
> > > >
> > > > ### Additional Remarks:
> > > >
> > > > We would like to highlight that we added five additional state-of-the-art networks to the main results in the manuscript as requested by other reviewers to modernize our baselines, culminating in about 225 new training runs, in case the reviewer is also interested in these values.
> > > >
> > > > ### References
> > > > [1]: Isensee, Fabian, et al. "nnu-net revisited: A call for rigorous validation in 3d medical image segmentation." International Conference on Medical Image Computing and Computer-Assisted Intervention. Cham: Springer Nature Switzerland, 2024.
> > > >
> > > > [2]   Brugnara, Gianluca, et al. "Deep-learning based detection of vessel occlusions on CT-angiography in patients with suspected acute ischemic stroke." Nature Communications 14.1 (2023): 4938.
> > > >
> > > > [3] Wong, Oi Yean, et al. "Hyperacute stroke pathway CT reporting times at UCLH." Clinical Radiology 72 (2017): S20.

---

### Author Response · Authors · 2025-11-25
**General Rebuttal Response**

Thank you to all the time and effort of the reviewers in reading and thoroughly evaluating our paper.  We received a lot of helpful comments, which we thoroughly considered and addressed in this rebuttal. We have uploaded a revised paper version, which displays all new additions in blue for easier visibility. Among the additions are:
1. An extended motivation on why strong transformers in 3D medical image segmentation are needed
2. Inclusion of 5 more recent 3d medical architectures, namely MedNeXt, UNETR++, SwinUNETR-V2, SegMamba, and VisionxLSTM.
3. An extended discussion in Section 5 clarifying that hybrid architectures are not problematic, but that the performance of the underlying transformer matters.
4. An additional discussion in Appendix F about the applicability of our proposed encoders for various clinical tasks with respect to inference time.
In our responses, we address the concerns of each reviewer individually, responding to their concerns. Due to the overlap of certain criticisms, responses may repeat between reviewers; hence, we recommend skimming over those sections should you read our response to other reviewers.

Lastly, we would like to thank all reviewers again for the input, as we thoroughly believe it helped improve the overall quality of our paper. Should your concerns be resolved and you agree with the improved quality of our revision, we hope you will raise the scores accordingly to reflect this.

Should any questions remain, we look forward to engaging in further discussion.

---

> ### Author Response · Authors · 2025-11-26
> **Update: All results for five new state-of-the-art baselines are now available**
>
> All of our pending training runs as noted in our comments yesterday have been completed and likewise updated in both the revised manuscript and in our rebuttal comments to the reviewers.

---

### Author Response · Authors · 2025-12-01
**Consolidated Response for Area Chair Review**

We recognize that the current situation places an unusual burden on ACs and that decisions are made under time pressure and without discussion. Hence, we provide a summary of the rebuttal to simplify the evaluation, especially since we believe the reviewer's concerns were fully addressed by the revision and that the leak prevented us from verifying this with the reviewers.

## 1. Why this paper matters
Transformers are the foundation of modern natural image modeling, self-supervised learning (SSL), and multimodal systems. Yet in 3D medical image segmentation, pure Transformers have repeatedly failed to be competitive. Reviewers noted this gap explicitly, i.e. n3eQ, who questioned why a Transformer-centric model is needed when CNNs still dominate.

Our work identifies, for the first time, why Transformers have been ineffective in this domain and what is required to make them succeed. As highlighted by J9TG, the analysis provides a clear and quantitative explanation: existing 3D Transformer-CNN hybrids only appear to succeed because their convolutions perform the actual representational work, while the Transformer blocks contribute little or nothing. This insight is scientifically important on its own, since the field could not progress without understanding this failure mode.

Building on that analysis, we introduce Primus and PrimusV2, the first architectures in which the Transformer truly carries the representational load. Vcda emphasized that our design choices address the unique challenges of 3D, and our ablations substantiate each step. With these changes, PrimusV2 becomes the first Transformer to reach parity with the strongest CNN baselines across large-scale and diverse benchmarks in 3D medical image segmentation. This closes a long-standing performance gap and provides a strong 3D medical image Transformer that reaches CNN parity, creating a solid foundation for future methodological advances and enabling the 3D medical domain to meaningfully engage with innovations emerging from Transformer research in natural images, especially in SSL.
## 2. How we addressed the key reviewer concerns
### A. Outdated baselines (J9TG, n3eQ, Dpm7)
We added MedNeXt, UNETR++, SwinUNETR-V2, SegMamba, and VisionxLSTM for all nine datasets. This required 225 new training runs, resulting in 11 baselines collectively. The updated Table 6 shows that PrimusV2 remains the top Transformer and matches the strongest CNNs. Full details are in the revised manuscript.
### B. Motivation for Transformers (n3eQ, Dpm7)
We clarified in the Introduction ("Why Transformers?" paragraph) that progress in representation learning, including DINO, MAE, iBOT, DinoV2, JEPA, V-JEPA, and DinoV3, have exclusively leveraged pure Transformer architectures. Since 3D medical imaging lacked a competitive Transformer, the field has been unable to adopt these major advances. PrimusV2 resolves this obstacle and restores alignment with the broader research ecosystem.
### C. Concerns about hybrid architectures (all reviewers)
We clarify that we do not criticize hybrids in general, but that earlier hybrids in 3D medical image segmentation hid the ineffectiveness of their Transformer components. The updated Discussion now states this explicitly. Our ablations show that removing the Transformer from prior hybrids barely changes performance, while removing it from Primus V1 and V2 leads to large drops. This demonstrates that our Transformer backbone is genuinely effective in driving segmentation performance.
### D. Model size concerns (Vcda, Dpm7)
We clarify that model size does not impact clinical feasibility (new Appendix F). We demonstrate using concrete examples that even the largest model fits comfortably within clinical time constraints for tasks such as emergency stroke assessment. Moreover, performance does not come from scale. This is evidenced by the fact that PrimusV2 Small, at 24.6M parameters, already surpasses nnU-Net and all previous Transformers.
### E. Why CNNs historically outperform Transformers (Dpm7)
This was indeed already part of our original manuscript which identifies the actual architectural barriers that lead to Transformer underperformance in 3D medical image segmentation: coarse tokens, unsuitable positional encodings, and weak tokenizers. Section 3.2, 3.3 and Table 3 demonstrates explicitly how removing each barrier systematically closes the performance gap.
## 3. Conclusion
We hope this short summary assists your evaluation given the unusual circumstances and limited opportunity for reviewer engagement. The paper identifies why Transformers have historically failed in 3D medical segmentation, demonstrates how to fix these limitations, and provides the first Transformer architecture that performs on par with the strongest CNNs. All reviewer concerns have been addressed directly in the revised manuscript (visualized in blue) and in our detailed responses. We appreciate your effort and look forward to a positive evaluation of our work.

---

### Meta-Review · Area_Chair_vm8N · 2026-01-12

**Summary:**

Reviewers raised several concerns with the manuscript including, inference latency and memory usage of the proposed model (e.g., higher parameters of the proposed Primus models compared to nnU-Net), empirical results suggesting the convolutional residual blocks can learn effective features and maintain high performance even without the transformer - contradicting the overall motivation of the manuscript, missing comparison with several recent and strong segmentation models (e.g., MedNeXt, UNETR++, SwinUNETR‑V2), questionable motivation with respect to existing hybrid CNN–Transformer models, and limited performance gains.

**Reviewer Concerns:**

The meta-reviewer believes some of the initially raised concerns are partially addressed by the rebuttal.  For instance, authors provided additional experimental comparison with MedNeXt, SwinUNETR-V2, UNETR++, SegMamba and VisionxLSTM. Authors also provided additional discussion with respect to inference time.

**Reviewer Scores:**

The meta-reviewer believes the rebuttal only partially addresses reviewer's concerns. For intance, the issue of questionable motivation and limited performance gains (e.g., comparison with ResEnc-L and MedNeXt) still remains. These issues are raised by multiple reviewers.  Majority (3) reviewers generally remained negative highlighting multiple and somewhat overlapping issues. Based on the reviewer's comments and rebuttal, the meta-reviewer believes the negative concerns are comprehensive and are not fully resolved in the rebuttal.

---

### Decision · Program_Chairs · 2026-01-26

Reject